# Impact of ice multiplication on the cloud electrification of a cold-season thunderstorm: a numerical case study

**Jing Yang**[1,2,★]**, Shiye Huang**[1,★]**, Tianqi Yang**[3]**, Qilin Zhang**[1]**, Yuting Deng**[1]**, and Yubao Liu**[1]

[1]Collaborative Innovation Center on Forecast and Evaluation of Meteorological Disasters (CIC-FEMD), China Meteorological Administration Aerosol-Cloud and Precipitation Key Laboratory, Precision Regional Earth Modeling and Information Center (PRMIC), Nanjing University of Information Science and Technology, Nanjing, 210044, China
[2]China Meteorological Administration Key Laboratory of Cloud-Precipitation Physics and Weather Modification (CPML),
Beijing, 100081, China
[3]Nanjing Meteorological Bureau, Nanjing, 210019, China
★These authors contributed equally to this work.

**Correspondence:** Yubao Liu (ybliu@nuist.edu.cn)

**Abstract.** Ice microphysics controls cloud electrification in thunderstorms, and the various secondary ice production (SIP) processes are vital in generating high ice concentrations. However, the role of SIP in cold-season thunderstorms is not well understood. In this study, the impacts of SIP on the electrification in a thunderstorm that occurred in late November are investigated using model simulations. The parameterizations of four SIP processes are implemented in the model, including the rime splintering, ice–ice collisional breakup, shattering of freezing drops, and sublimational breakup of ice. In addition, a noninductive charging parameterization and an inductive charging parameterization, as well as a bulk discharging model, are coupled with the spectral bin microphysics scheme. The macroscopic characteristics and the temporal evolution of this thunderstorm are well modeled. The radar reflectivity and flash rate obtained by adding four SIP processes are more consistent with the observations than those without SIP. Among the four SIP processes, the rime splintering has the strongest impact on the storm. The graupel and snow concentrations are enhanced while their sizes are suppressed due to the SIP. The changes in the ice microphysics result in substantial changes in the charge structure. The total charge density changes from an inverted tripole structure to a dipole structure (tripole structure at some locations) after four SIP processes are considered in the model, mainly due to the enhanced collision between graupel and ice. These changes lead to an enhancement of the vertical electric field, especially in the mature stage, which explains the improved modeling of flash rate. The results highlight that cold-season cloud electrification is very sensitive to the SIP processes.

## 1 Introduction

Cold-season thunderstorms may have different characteristics in terms of charge structure and lightning activity compared to warm-season thunderstorms due to the different thermodynamic conditions (Michimoto, 1991; Takahashi et al., 1999; Caicedo et al., 2018). Caicedo et al. (2018) investigated the differences between cold-season and warm-season thunderstorms in north-central Florida using the Lightning Mapping Array (LMA) and radar data. They showed an apparent discrepancy in that all the observed charge areas of the summer storms were located up to 1 km higher than in winter and spring storms; this was also the case for the 0, $-10$, and $-20\,°C$ isotherms. The average LMA

initiation power in winter and spring storms was about 1 order larger than in summer storms. This result is supported by the electric-field measurements of the initial breakdown process by Brook (1992), who assured that cloud-to-ground discharges and intracloud discharges were probably more energetic in winter than in summer. Wang et al. (2021) reported that, in contrast to lightning in summer, which mostly delivered negative charges to the ground, 30 % of cloud-to-ground lightning in Honshu Island winter thunderstorms delivered positive charges to the ground. They attributed this phenomenon to inverted charge structures. The apparent differences between the cold-season and warm-season thunderstorms indicate different characteristics of ice microphysics that control cloud electrification.

Extensive studies have been conducted to understand the role of ice microphysics in cloud electrification in summertime thunderstorms (e.g., Mansell et al., 2010; Fierro et al., 2013; Qie et al., 2015; Qie and Zhang, 2019; Zhang et al., 2016; Lyu et al., 2023), while fewer have been performed focusing on cold-season thunderclouds. Michimoto (1991) investigated the behavior of both 30 and 20 dBZ radar echoes in early winter thunderstorms and found that lightning occurred as 30 dBZ radar echoes reached $-20\,^{\circ}\mathrm{C}$, from which it could be inferred that lightning was related to the interaction of graupel and ice crystals. Zheng et al. (2019) analyzed the charge distribution of cells in three winter thunderstorms in the Hokuriku region of Japan based on LMA and radar data. They suggested that riming electrification between graupel and ice crystals or their aggregations are the dominant mechanisms for the electrification in most cells, and the charging process between snow aggregates is responsible for inverted charge structures that occur above the $0\,^{\circ}\mathrm{C}$ isotherm. Using a variety of observational data from Videosondes and Videosonde-HYVIS conjoined sondes, radar, and the Lightning Location System Network, Takahashi et al. (2019) revealed that the frequent lightning activity produced by shallow winter thunderclouds in Hokuriku is probably due to the high number concentration of ice crystals.

One of the key mechanisms of ice generation in deep convective clouds is ice multiplication, i.e., secondary ice production (SIP), which refers to the ice fragments produced during the interactions between different hydrometeors or the freezing of supercooled drops. SIP is the main explanation for why the observed ice concentration is orders of magnitude higher than the ice nucleating particles (INPs; Hallett and Mossop, 1974; Heymsfield and Willis, 2014; Yang et al., 2016; Korolev and Leisner, 2020). Some studies have tried to investigate the impact of SIP on cloud electrification in summer (e.g., Fierro et al., 2013; Latham et al., 2004; Mansell et al., 2010; Phillips et al., 2020; Phillips and Patade, 2022), mostly based on numerical simulation since a limitation of observation is that it can hardly separate different ice generation processes. For example, Latham et al. (2004) investigated the role of the rime-splintering process in lightning activity using model simulation; they suggested that

the relationship between flash rate and precipitation intensity is linear if not considering SIP, while this relationship changed to being nonlinear with the SIP included. However, rime splintering is not the only SIP process that can influence the charge structure of thunderstorms. Secondary ice can be produced through various processes, such as the shattering of freezing drops, ice–ice collisional breakup, and sublimational breakup of ice (Lauber et al., 2018; Phillips et al., 2018; Korolev and Leisner, 2020; Deshmukh et al., 2022). Recently, Phillips and Patade (2022) showed that the ice–ice collisional breakup may significantly alter the charge structure of summertime thunderstorms using a high-resolution cloud model.

Till now, to our best knowledge, no study has investigated the role of different SIP processes in cloud electrification under cold-season conditions using numerical simulations. However, there are a few modeling studies that highlighted the importance of ice generation in wintertime cloud electrification. For example, Takahashi (1983) studied electrical development in winter thunderclouds using an axisymmetric cloud model. The results showed that no strong electrification was observed before the appearance of the solids, which implies the importance of the riming charging for the electrification. Thus, the generation of graupel perhaps plays a vital role in wintertime cloud electrification, while SIP controls the fast graupel generation in convective clouds (Yang et al., 2016; Takahashi et al., 2019). Using the Regional Atmospheric Modelling System (RAMS) mesoscale forecast model, Altaratz et al. (2005) analyzed the charge separation in winter convections using different parameterizations of a noninductive charging mechanism, and they showed that the charge structure is very sensitive to the choice of ice microphysics scheme.

In this study, we performed a real-case simulation using the Weather Research and Forecast (WRF) model coupled with a spectral bin microphysics (SBM) scheme (Khain et al., 2004) and a bulk lightning model (Fierro et al., 2013) to investigate the impacts of SIP on cold-season thunderstorms. Parameterizations of four different SIP processes and an inductive and a noninductive charging parameterization (Saunders and Peck, 1998; Mansell et al., 2005, 2010) are implemented in the fast-SBM scheme. The SIP processes considered here include rime splintering, ice–ice collisional breakup, shattering of freezing drops, and sublimational breakup of ice. The rest of the paper is organized as follows: Sect. 2 describes the model and design of numerical experiments. Section 3 shows the results, including the model validation and the impacts of different SIP processes on cloud microphysics and charge structure. A discussion and conclusions are presented in Sect. 4. The parameterizations used in this study are detailed in Appendix A and B.

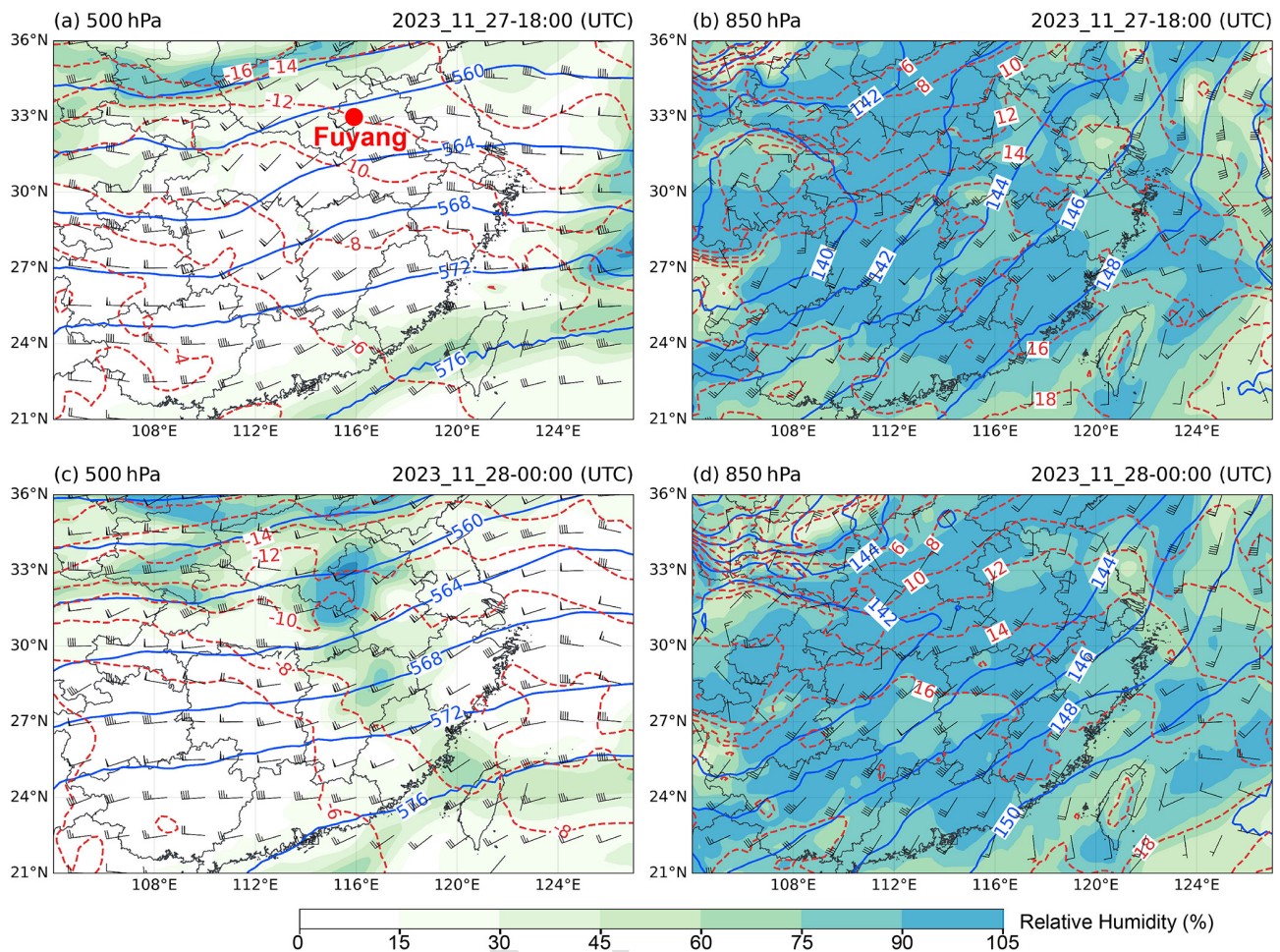

**Figure 1.** Synoptic conditions of the thunderstorm that occurred at **(a, b)** 18:00 UTC on 27 November and **(c, d)** 00:00 on 28 November. Panels **(a)** and **(c)** show the 500 mb geopotential height, isotherms, and wind barbs. **(b, d)** Same as **(a)** and **(c)** but for 850 mb. The red dot in **(a)** indicates the location of the sounding measurement that is shown in Fig. 2.

## 2 Model description and design of numerical experiments

### 2.1 Case description

On 27–28 November 2022, a severe thunderstorm occurred in southeastern China. The storm began at about 15:00 UTC (unless otherwise noted, UTC time is used throughout the remainder of this paper) on 27 November and lasted for more than 18 h. Figure 1 shows the synoptic conditions at 18:00 on 27 November and at 00:00 on 28 November, plotted using the fifth-generation ECMWF reanalysis (ERA5) data. At 500 hPa, the relative humidity was low in southeastern China at 18:00 on 27 November (Fig. 1a). Westerly wind prevailed, and the temperature ranged from $-6$ to $-12\,°C$. A weak short wave was present between 108 and $112°E$ and was moving towards the east. At 850 hPa (Fig. 1b), the southwesterly wind brought warm moist air to southeastern China, and the low-level relative humidity was very high, resulting in a nearly saturated condition. Baroclinicity was present, as seen

from the wind blowing across the isotherms. The moist low-level and dry high-level conditions are favorable for convection formation. At 00:00 on 28 November, two areas with relatively high relative humidity were observed at 500 hPa, especially near Fuyang, where the air was saturated. This is because two convective cells had already formed at this time. The low-level southwesterly wind kept providing warm moist air during the development of the convection.

The synoptic condition is also evident in the sounding measurement. As seen in Fig. 2, at 12:00 on 27 November, there was a deep moist layer from the surface up to 700 hPa, and the specific humidity decreased substantially above 700 hPa. The low-level wind was southwesterly, and the upper-level wind was westerly. Due to the southwesterly warm air, the temperature near surfaces was approximately $18\,°C$, which is higher than the typical temperature in November in this region but is about $10\,°C$ lower than that in summer. Potential instability was present in such a thermodynamic environment, providing favorable conditions for

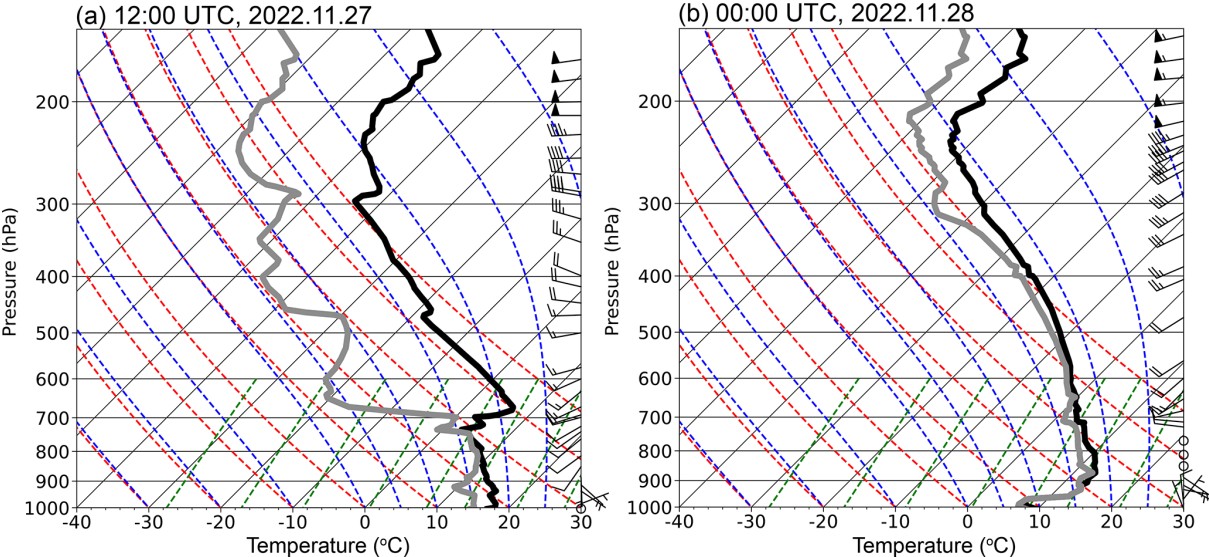

**Figure 2.** Skew-T log-p diagrams of sounding data from Fuyang at 12:00 UTC on 27 November and 00:00 UTC on 28 November 2022. The black profiles indicate the temperature, and the gray profiles indicate the dew point.

deep convection to occur. At 00:00 on 28 November, the air was nearly saturated below 500 hPa as the convective clouds had formed. There was an inversion layer near the surface, probably due to the cold pool induced by the convective pre-cipitation.

The radar composite reflectivity at different times in southeastern China is shown in Fig. 4g–i. At 02:00 on 28 November, two deep convective clouds were observed, extending from southwest to northeast and generating lightning flashes (Fig. 5a). The reflectivity in the convective core was approximately 50 dBZ. The entire system moved towards the east, and the east convective cloud moved to the sea after 06:00 (Fig. 4i). The intensity of the storm remained similar between 02:00 and 06:00, while the scale of these two convections increased slightly during the eastward propagation. The storm left the continent and continued on the sea after 08:00 on 28 November (not shown).

## 2.2 Model setup and design of numerical experiments

In this simulation, a two-way nested domain is used (Fig. 3). The outer domain has a grid spacing of 9 km. The grid spacing of the inner domain is 3 km, with $328 \times 298$ grids. There are 51 vertical levels, with a top pressure of 50 hPa ($\sim 20$ km). The ERA5 reanalysis data, which have a horizontal resolution of $0.25° \times 0.25°$ and an hourly temporal resolution, are used to drive the model and provide the boundary condition. The simulation runs from 12:00 on 27 November to 12:00 on 28 November, with a spin-up time of 12 h. The fast version of the SBM scheme is used to model the cloud microphysics. Compared to the bulk microphysics scheme, the SBM scheme has the advantage of calculating particle size distributions (PSDs) by solving explicit microphysical

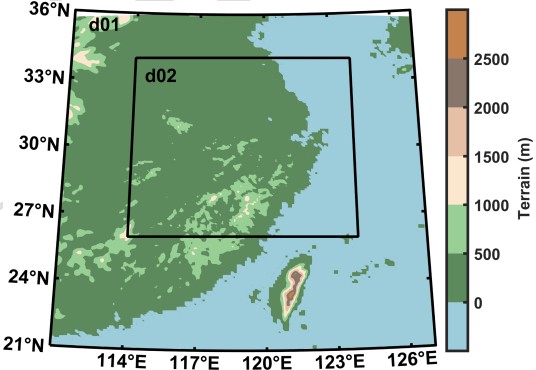

**Figure 3.** Domains of WRF model simulation.

equations. It aims to simulate the cloud microphysical processes as accurately as possible (Khain et al., 2015). In the fast version of SBM in WRF, the ice and liquid hydrometeor species include cloud droplets or rain, ice or snow, and graupel; each of them is represented by 33 doubling mass bins. It has been demonstrated in many previous studies that SBM performs better than bulk microphysics in modeling cloud microphysics (e.g., Fan et al., 2012; Khain et al., 2015). However, SBM has not been widely used for studying cloud electrification (e.g., Mansell et al., 2005; Shi et al., 2015). Recently, Phillips et al. (2020) implemented the cloud electrification parameterization in the SBM in a cloud model, and they conducted an idealized simulation of a deep convective cloud. The results showed that the modeled charge structure and lightning activity are consistent with observations. However, cloud electrification has not been implemented in SBM in WRF for a real case study before.

The Kain–Fritsch cumulus scheme is used for the outer domain, while it is turned off for the inner domain. The other physical choices include the rapid radiative transfer model for shortwave and longwave radiation (Mlawer et al., 1997), the revised MM5 surface layer scheme (Jiménez et al., 2012), the Noah land surface model (Tewari et al., 2004), and the Yonsei University planetary boundary layer scheme (Hong et al., 2006).

Parameterizations of four SIP mechanisms are implemented in the SBM: the rime splintering, ice–ice collisional breakup, shattering of freezing drops, and sublimational breakup of ice. Their equations are detailed in Appendix A. The parameterization of rime splintering is developed based on the laboratory experiments conducted by Hallett and Mossop (1974), which showed that an ice splinter is created for every 200 droplets collected by a graupel through riming at −5 °C. This SIP rate decreases as the temperature increases or decreases from −5 °C. At temperatures colder than −8 °C or warmer than −3 °C, the rime splintering is inactive. The parameterization of the shattering of freezing drops is also developed based on previous laboratory experiments (King and Fletcher, 1973; Phillips et al., 2018). It is a set of functions depending on the particle size and temperature. In this mechanism, either tiny or big ice fragments can be produced when a supercooled liquid drop collides with an ice crystal. The production rate of ice fragments is the highest at −15 °C, but it can also be active at colder and warmer temperatures (Lauber et al., 2018). The parameterization of ice–ice collisional breakup is developed based on the principle of energy conservation, as well as previous laboratory experiments (Takahashi et al., 1995; Yano and Phillips, 2011; Phillips et al., 2017). The production rate depends on the density and shape of ice particles, as well as the collision kinetic energy. Deshmukh et al. (2022) proposed a formulation for the number of ice splinters generated during ice sublimation based on laboratory observations. The relative humidity on the ice and the preliminary size of the mother ice particles both govern the number of ice splinters. The formulation is used for dendritic crystals and heavily rimed particles (e.g., graupel). Waman et al. (2022) simulated a squall line with four SIP processes and found that sublimation fragmentation is only active in downdrafts.

Similarly to many previous studies (Mansell et al., 2010; Fierro et al., 2013; Guo et al., 2017), we use the parameterization of noninductive charging developed by Saunders and Peck (1998) to simulate the cloud electrification, which is a function of particle terminal velocity, collisional efficiency, temperature, and rime accretion rate (RAR). This parameterization is supported by a series of laboratory experiments demonstrating that collision between graupel and ice is the key noninductive charging mechanism (e.g., Brooks et al., 1997; Takahashi and Miyawaki, 2002; Saunders and Peck, 1998; Saunders et al., 2001; Emersic and Saunders, 2010). Some modeling studies showed this parameterization would result in an inverted charge structure (e.g., Mansell et al., 2010; Phillips et al., 2020) in a thunderstorm, while in this study, we will show that, with SIP implemented in the model, the charge structure changes from inverted to normal, suggesting that the correct representation of ice generation is vital in modeling the cloud electrification. In addition, a parameterization of inductive charging (Mansell et al., 2005) is implemented in the SBM. The charge transfer occurs during the riming process between polarized supercooled droplets and graupel along grazing trajectories (Moore, 1975). With charge density modeled, the electric field can be calculated based on the Poisson equation, and the discharge is simulated using a bulk model (Fierro et al., 2013). The equations of these parameterizations can be found in Appendix B.

Six sensitivity experiments are designed to investigate the impacts of different SIP processes on cloud electrification. In the first experiment, none of the SIP parameterizations are used (hereafter noSIP); in the second experiment, only rime splintering is considered (hereafter RS); in the third experiment, only ice–ice collisional breakup is used (hereafter IC); in the fourth experiment, only shattering of freezing drops is turned on (hereafter SD); in the fifth experiment, only sublimational breakup of ice is applied (hereafter SK); in the last experiment, all four of the SIP mechanisms are considered (hereafter 4SIP).

## 2.3 Description of observation dataset

Radar reflectivity can be used to illustrate the intensity of the storm. The radar data used in this study constitute a gridded product generated based on 32 S-band radars operated across southeastern China. For each radar, the detection radius is 230 km, the range resolution is 250 m, and the beamwidth is 1°. The radar finishes a volume scan every 6 min, consisting of nine elevation angles (0.5, 1.5, 2.4, 3.4, 4.3, 6.0, 9.9, 14.6, and 19.5°). The data recorded by these radars were interpolated into a Cartesian grid with a horizontal resolution of 1 km and a vertical resolution of 500 m based on the Cressman technique.

In addition, the lightning location and flash rate are evaluated using observation. The lightning location data are obtained based on the very-low-frequency (VLF) lightning location network (LLN) in China, developed by Nanjing University of Information Science and Technology (Li et al., 2022). The VLF-LLN was established in 2021 and has 26 stations distributed across various regions in China. The detection area covers the entirety of China, as well as parts of East and Southeast Asia. The lightning location algorithm is developed based on the time-of-arrival (TOA) method, and the arrival times of each lighting-induced pulse at different stations are obtained by matching the recorded waveforms to the idealized waveforms simulated using the finite-difference time-domain (FDTD) technique. The lightning location error is 1–5 km (Li et al., 2022).

Moreover, the ERA5 reanalysis data are used to investigate the synoptic conditions; the sounding measurement at

Fuyang, which is conducted every 12 h, is used to investigate the thermodynamic conditions; and the brightness temperature (TBB) on the FY2H satellite that is developed in China is used to illustrate the cloud coverage.

## 3   Results

### 3.1   Model evaluation

The composite radar reflectivity modeled in the noSIP and 4SIP numerical experiments is compared with the observation in Fig. 4. It is within the expectation that the simulated convection inevitably deviates from that which is observed (Fig. 4g–i) to some extent, but, in general, the model captures well the location and scale of the storm. The model also successfully simulates the east propagation of the storm (Fig. 4a–c). The SIP processes have minor impacts on the macro-properties of the storm, while the intensity can be clearly affected. At 02:00 on 28 November, the noSIP experiment overestimates the composite radar reflectivity; the modeled area with reflectivity greater than 45 dBZ is much larger than observed (Fig. 4a and g). With all four SIP processes implemented, the simulation result is more consistent with the observation (Fig. 4d–f) and is better than the experiments with a single SIP process (not shown). Similarly, at 04:00 and 06:00, the radar reflectivity is overestimated in the noSIP experiment (Fig. 4b and c). With all four SIP processes considered together, the simulation result is more consistent with the observation than that without SIP, not only for the intensity but also for the shape of the east convective cloud (Fig. 4d–f).

To statistically investigate the difference in the reflectivity at different heights between observations and model simulations, the contoured-frequency-by-altitude diagrams (CFADs) of reflectivity are plotted (Fig. 5). As seen in Fig. 5, the maximum reflectivity is observed at about 4 km (Fig. 5g–i), which is the height of the melting level. The modeled maximum reflectivity from the noSIP experiment (Fig. 5a–c) is larger than observed by about 7 dBZ; this is also seen from the map of composite reflectivity in Fig. 4. With SIP implemented, the maximum reflectivity decreases and is more consistent with observations (Fig. 5d–f). Since the radar reflectivity is calculated for a wavelength of 10 cm, which is more sensitive to particle size, the decreased reflectivity implies smaller particle sizes after SIP processes are used in the model; this will be demonstrated in Sect. 3.2. The mean reflectivity profiles in both the noSIP and 4SIP experiments are systematically larger than observed as the occurrence frequency of reflectivity greater than 30 dBZ is overestimated, but the 4SIP experiment performs better than the noSIP experiment. Note that the observed reflectivity is underestimated at low levels because the lowest elevation angle used in the radar measurement is 0.5°, and the low-elevation beams are affected by ground clutters. Based on the facts that composite reflectivity is simulated reasonably well and that

the SIP processes result in improvements, we are confident in investigating the impacts of SIP on the cloud microphysics and electrification in the cold-season storm.

The lightning locations and flash rates from the observations and the numerical experiments are compared in Fig. 6. Since we use a bulk discharge model in simulating the flash, it is within expectation that there are uncertainties in modeling the lightning frequency. In addition, the lighting occurrence is strongly related to the convective cores; the uncertainty in modeling the flash rate is associated with the uncertainty in modeling the radar reflectivity (Figs. 4 and 5). It is seen from Fig. 6a that the lightning locations obtained from the simulations are in agreement with the observations in the southern convection. The simulated lightning locations are in the low TBB (brightness temperature) region, which implies strong convection. The number of lightning flashes obtained from the simulation in the northern cell (29–32° E) is much less than observed as WRF failed to simulate the deep convection. The temporal evolution of flash rate in the southern convection is shown in Fig. 6b; it is seen that there is improvement in modeling the temporal variation of flash rate by implementing SIP processes. The observation indicates that the highest flash rate occurred between 00:00 and 01:00 on 28 November. Without any SIP, the flash rate is relatively high before 00:00 on 28 November. The ice–ice collisional breakup enhances the flash rate and peaks at about 00:00 on 28 November. The flash rate has a similar magnitude in the noSIP and IC experiments. The rime splintering and shattering of freezing droplets can improve the simulation as the modeled flash rate is enhanced after 00:00 on 28 November, which is more consistent with observations. The simulated flash rate in the SK experiment peaks at 00:00 on 28 November, with a similar magnitude compared to that in the IC experiment. With all implemented, the modeled result is more consistent with the observations than the other experiments after 00:00. Overall, WRF captures the lightning locations and the temporal evolution of flash rate; this provides the basis for further analyzing cloud electrification.

### 3.2   The impact of ice multiplication on cloud microphysics

The various SIP processes may have different impacts on the cloud microphysics. Figure 7 presents the time–height diagrams of the mixing ratio and number concentration of graupel and/or hail, ice and/or snow, rain, and cloud water in the noSIP experiment. It is seen from the figure that the modeled convection was weak before 18:00 on 27 November, and only warm rain was present. After 20:00 on 27 November, the modeled cloud top reached approximately 12 km above the mean sea level (a.m.s.l.), and significant homogeneous ice production took place near $-40\,°C$ (Fig. 7f). Between 00:00 and 06:00 on 28 November, the surface rain was relatively strong, and the maximum graupel and rain mixing ratios were about 0.11 and 0.13 g kg$^{-1}$. The snow mixing ratio was

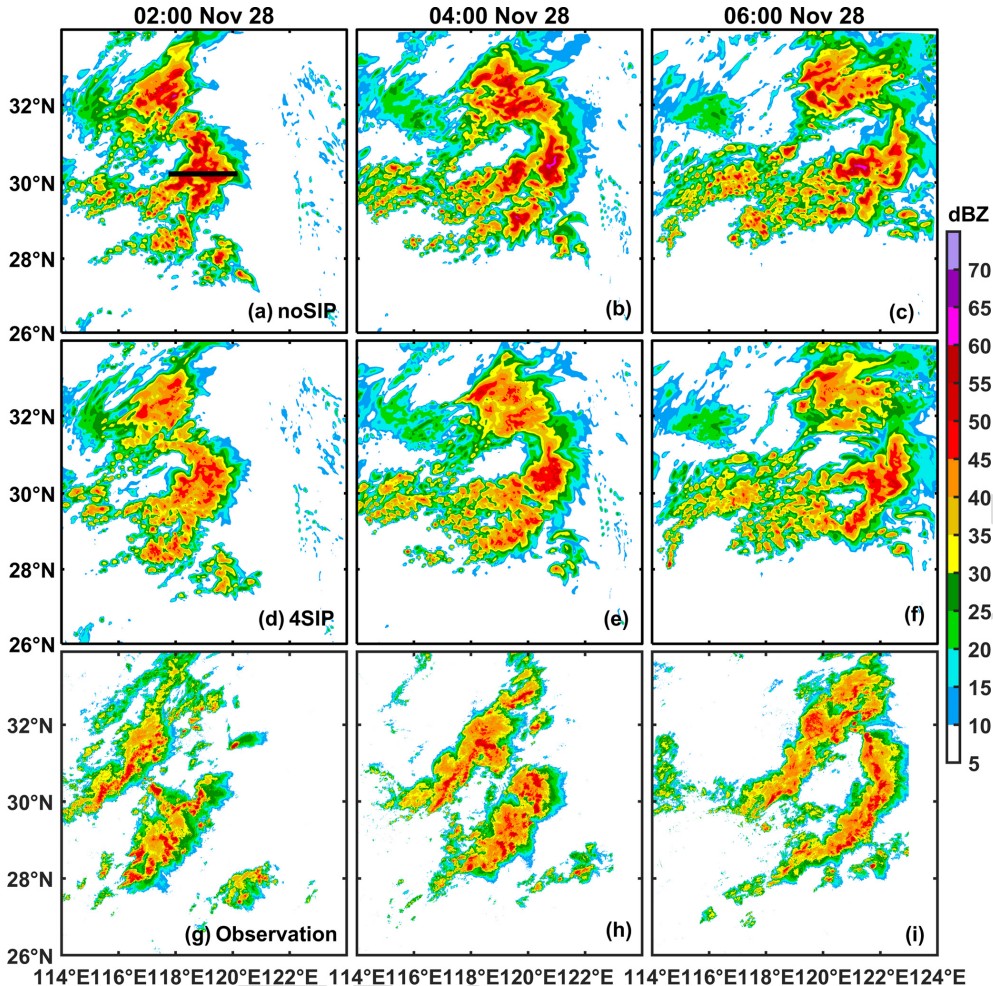

**Figure 4.** Composite radar reflectivity from **(a–c)** noSIP; **(d–f)** 4SIP experiment; and **(g–i)** observations at 02:00, 04:00, and 06:00 on 28 November. The horizontal black line in **(a)** shows the cross-section used in the following analysis.

higher than that of graupel and rain in this period. The temporal evolution of the rain mixing ratio is consistent with that of snow, suggesting that the melting of snow contributes significantly to the rain. After 01:00 on 28 November, the cloud top decreased, the surface rain was weakened, and the graupel and liquid-water mixing ratio decreased (Fig. 7a and d), suggesting a weakening of convection, and this resulted in the declining flash rate after 01:00 (Fig. 6).

The differences in the mixing ratios and number concentrations between the experiments with a single SIP process and four SIP processes and the noSIP experiment are shown in Figs. 8 and 9, respectively. As shown in Fig. 8a and f, the rime-splintering process has an enhancing effect on the graupel and the ice and/or snow mixing ratios throughout the cloud life cycle, mainly between 0 and $-20\,°C$. The maximum increase, which exceeds $0.02\,\mathrm{g\,kg^{-1}}$, is found between 00:00 and 04:00. However, the mixing ratios of rain and cloud droplets show a decrease above $0\,°C$, indicating the consumption of liquid water by the secondary ice

produced through the rime-splintering process. Thus, fewer cloud drops may be transported vertically to upper levels for freezing. The shattering of freezing drops also enhances the graupel and/or hail and the ice and/or snow mixing ratios (Fig. 8c and h) compared to noSIP. The enhancement of graupel occurs mainly between 0 and $-10\,°C$ and that of ice and/or snow occurs at a wider temperature range from 0 to $-40\,°C$. In addition, the liquid-water mixing ratio is reduced above the freezing level. The ice–ice collisional breakup and sublimational breakup of ice enhance the graupel mixing ratio and concentration after 02:00 on 28 November. Before 00:00, the ice concentration is high above $-30\,°C$, but the sizes of ice are all small; thus, the collisional breakup is insignificant. With all implemented, the graupel and snow mixing ratios and concentrations are enhanced throughout the cloud life cycle (Figs. 8e, j and 9e, j). The rime splintering and shattering of freezing drops are responsible for the enhancement of graupel and ice concentrations at 0 and $-30\,°C$ (Fig. 9e and j), and the ice–ice collisional breakup,

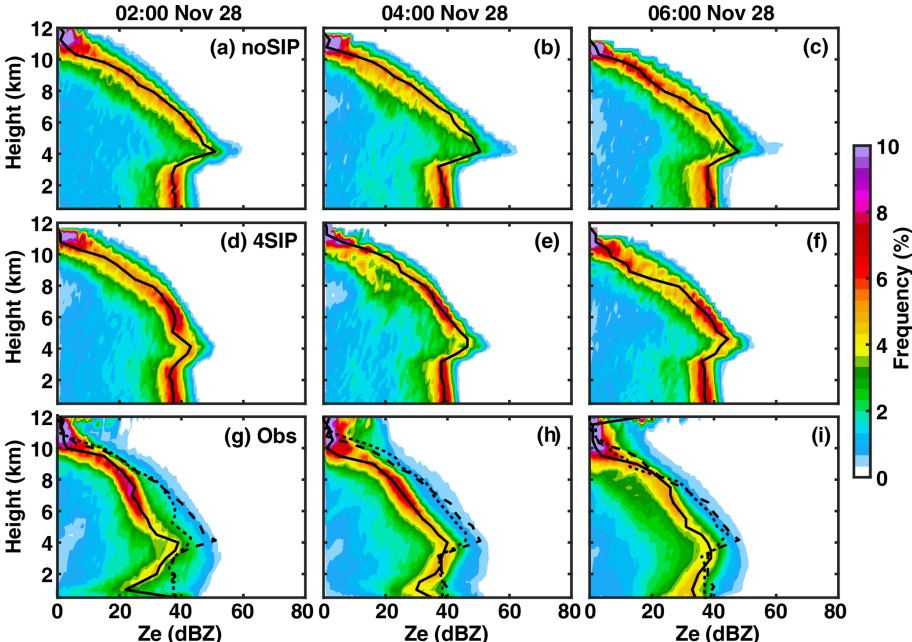

**Figure 5.** The CFAD of reflectivity from **(a–c)** noSIP and **(d–f)** 4SIP experiments and **(g–i)** radar observations at 02:00, 04:00, and 06:00 on 28 November. The black lines indicate the profiles of mean reflectivity, and the dashed and dotted lines in **(g–i)** are the mean reflectivity profiles from the noSIP and 4SIP experiments.

shattering of freezing drops, and sublimational breakup of ice are responsible for the ice concentration enhancement above $-40\,°C$ (Fig. 9j).

The enhanced graupel and/or hail and ice and/or snow con-
5 centrations and the decreased composite reflectivity by SIP processes imply decreased diameters of graupel and/or hail and ice and/or snow (Fig. 10). At temperatures warmer than $-20\,°C$, the graupel and/or hail and ice and/or snow sizes obtained from the RS experiments decrease by about 0.2 and
10 0.6 mm, respectively. In the region colder than $-20\,°C$, there is a slight increase in graupel size for both experiments, but the ice concentration remains similar after implementing SIP. The graupel and snow sizes are also reduced due to the shattering of freezing drops, and this decrease intensifies with de-
15 creasing height. On average, the ice–ice collisional breakup and sublimational breakup of ice have minor impacts on the graupel and ice size, which may be a result of the cancellation of regions with positive and negative impacts.

To understand the relative importance of the four SIP pro-
20 cesses, their ice production rates in the 4SIP experiment are illustrated in Fig. 11, which presents well the fact that the magnitudes and locations of secondary ice production are different among the four processes. As seen in Fig. 11a–d, the rime splintering and drop shattering produce significant sec-
25 ondary ice in the core of clouds, where the graupel and rain mixing ratios are high, while the sublimational breakup of ice is more intense near cloud edges or regions with relatively low reflectivity, probably because of the entrainment mixing and regional downdrafts. Ice–ice collisional breakup is more

intense in regions with high ice and/or snow concentrations;
its secondary ice production rate is much smaller than that of rime splintering. However, it should be noted that the efficiency of ice–ice collisional breakup is related to the rimed fraction (Karalis et al., 2022; Sotiropoulou et al., 2021). A sensitivity test shows that using a larger rimed fraction (0.4)
can result in a stronger impact of ice–ice collisional breakup on cloud microphysics, but it is still much weaker than that of rime splintering (not shown). The ice production rate by rime splintering is the highest, and that by the sublimational breakup of ice is the lowest. This substantial difference in the
magnitude of the ice production rate is also true after averaging the entire cloud region (Fig. 11e), and it explains why the rime splintering process has the most significant impact on the cloud microphysics on average.

### 3.3 The impact of ice multiplication on cloud electrification

The enhanced graupel and ice mixing ratios and concentrations may affect the charging rate by enhancing the graupel–ice collision and riming process. Figure 12 shows the average noninductive and inductive charging rate obtained from
50 the six numerical experiments. Note that the charging rate averaged over the cloud area is very small, and the maximum charging rate (not shown) is more than 4 orders of magnitudes larger than the average value, but the pattern is similar, thus providing the same conclusions. It is seen from
55 the figure that the cloud electrification starts at about 19:00

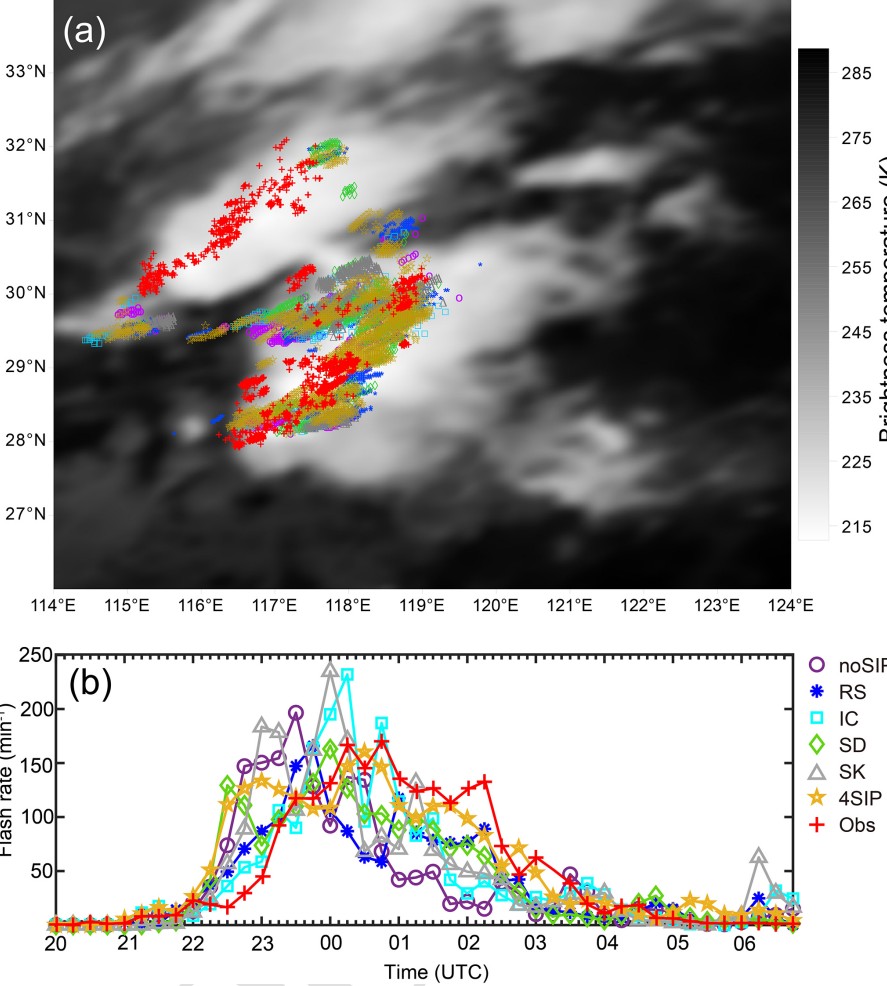

**Figure 6. (a)** The location of simulated and observed flashes over TBB and **(b)** the temporal variation of the simulated and observed flash rates.

on 27 November. Without any SIP considered in the model, the noninductive charging rate has an obvious separation at $-20\,°C$, with negative charging above this level and positive charging below (Fig. 12a). The magnitude of the upper-level negative charging rate is slightly larger than the positive charging rate.

However, with rime splintering included, the positive charging rate below 7 km is enhanced (Fig. 12b) as rime splintering is efficient at relatively warm temperatures. In fact, the rime-splintering process is mainly efficient between $-3$ and $-8\,°C$, but the secondary ice can be transported to higher levels in convection. The shattering of freezing drops also enhances the positive charging rate below 7 km (Fig. 12d). The ice–ice collisional breakup and sublimational breakup of ice only have weak impacts on the noninductive charging rate. With all four SIP processes included, the low-level positive noninductive charging rate on graupel is enhanced (Fig. 12k), mainly due to the composite impact of rime splintering and the shattering of freezing drops. The

magnitude of the upper-level negative noninductive charging rate remains similar compared to that without SIP.

The inductive charging rate is a few times smaller than the noninductive charging rate but cannot be neglected. The rime splintering and shattering of freezing drops result in very different structures of the inductive charging rate compared to that without SIP (Fig. 12g, h, and j). The upper-level negative charging on graupel in the noSIP experiment is changed to positive; this implies that the total charge structure may be inverted above 6 km due to these two SIP processes, which will be demonstrated later. In contrast, the distributions of the inductive charging rate in the IC and SK experiments are similar to that in the noSIP simulation. With all four SIP processes implemented, the inductive charging on graupel is positive at most of the levels (Fig. 12l), while at about $-10\,°C$, the graupel sometimes gets negative charging. This indicates the opposite sign of a vertical electric field; thus, positive-charge regions (or relatively weak negative-charge regions) are present at some locations at this level.

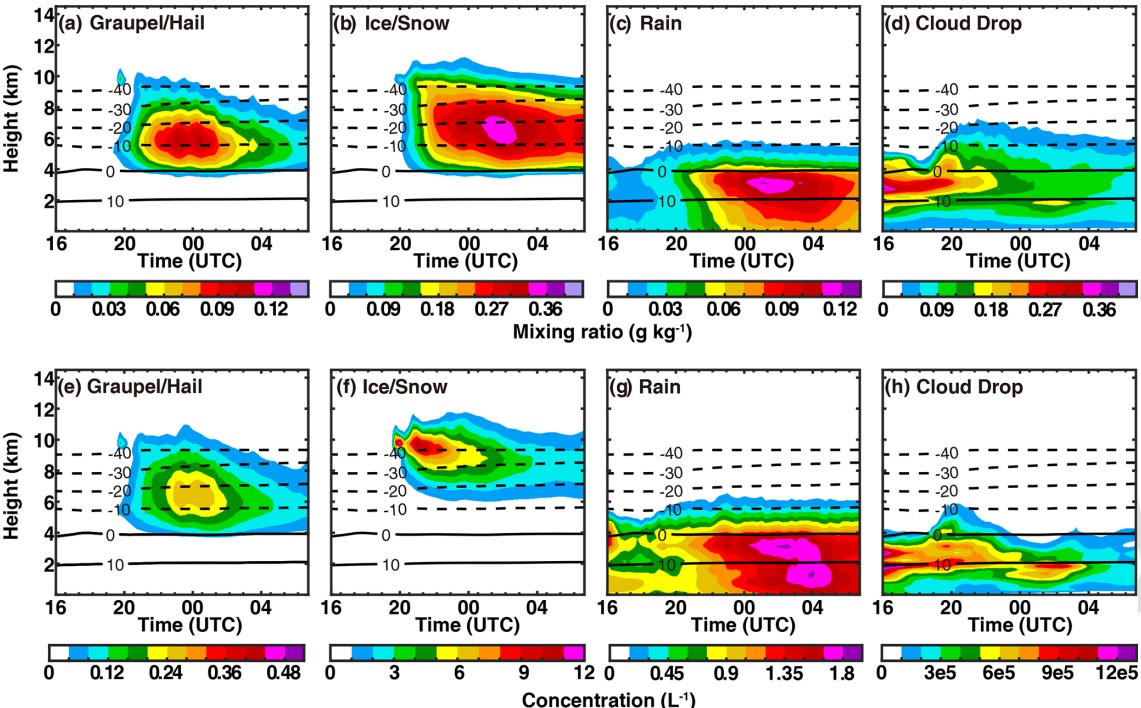

**Figure 7.** Time–height diagrams of **(a, e)** graupel and/or hail, **(b, f)** ice and/or snow, **(c, g)** rain, and **(d, h)** cloud droplet mixing ratio **(a–d)** and concentration **(e–h)** in the noSIP experiment.

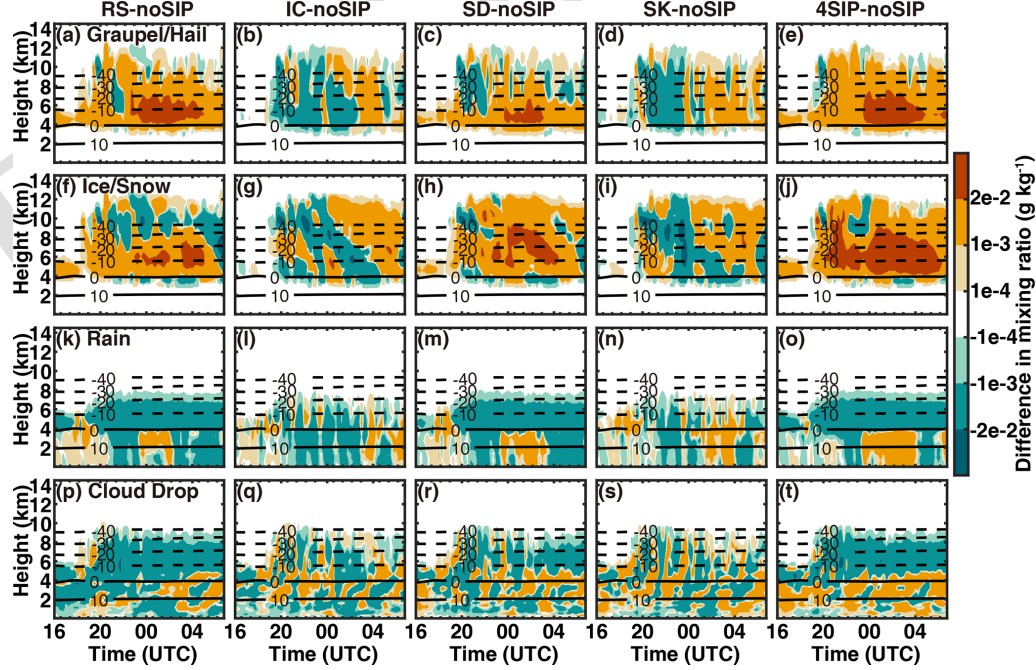

**Figure 8.** Differences in the mixing ratio of different hydrometeors between the experiments with SIP and those without SIP. **(a, f, k, p)** Experiment with rime splintering, **(b, g, l, q)** experiment with ice–ice collisional breakup, **(c, h, m, r)** experiment with shattering of freezing drops, **(d, i, n, s)** experiment with ice breakup during sublimation, and **(e, j, o, t)** experiment with four SIP processes.

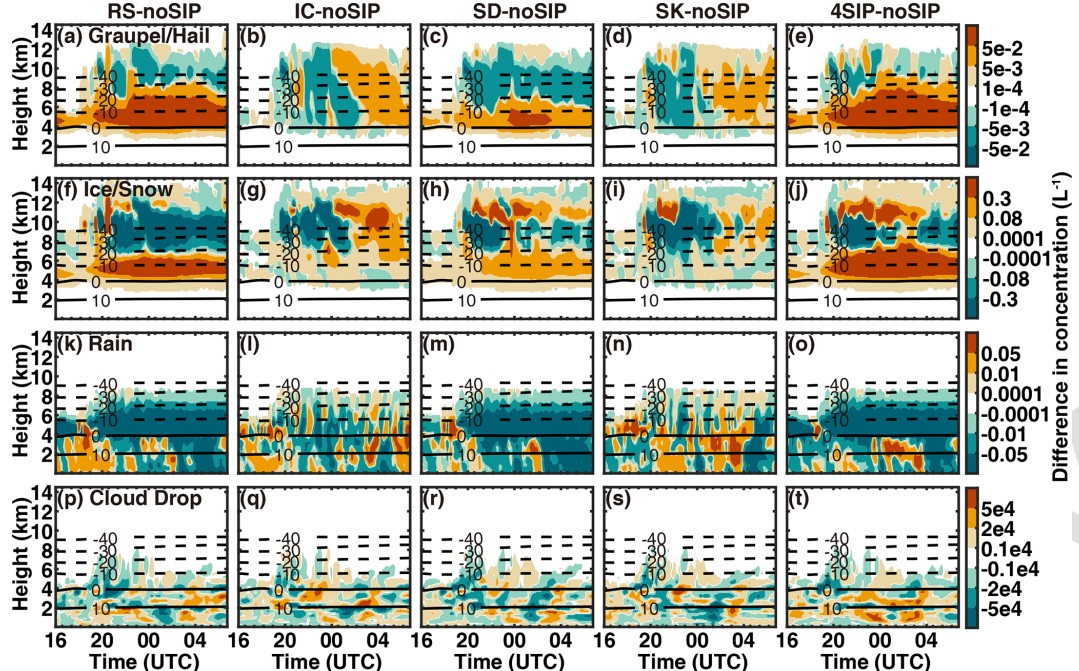

**Figure 9.** The same as Fig. 8 but for number concentration.

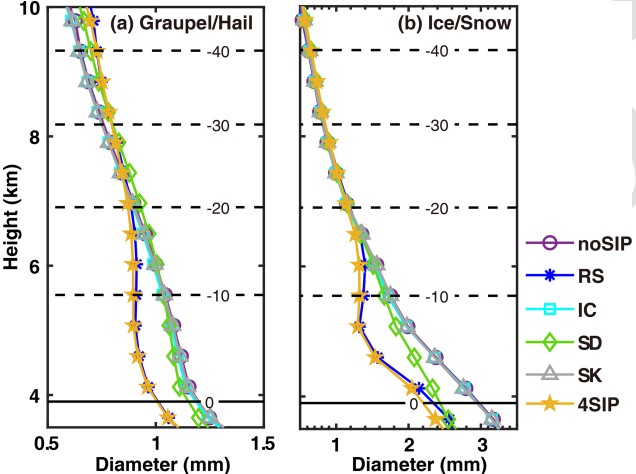

**Figure 10.** Profiles of the diameters of **(a)** graupel and **(b)** ice and/or snow.

The modified charging rate by SIP results in changes in the structure of charge density carried by different hydrometeors, especially the graupel and ice. As shown in Fig. 13, the average charge density carried by graupel and/or hail is negative at all levels when not considering the SIP. Although the graupel gets positive charge by colliding with ice below 8 km (Fig. 12a), the graupel falling from the upper levels brings a negative charge to the lower levels, resulting in the negative-charge density on average; this will be discussed in more detail in Figs. 14 and 15. In addition, the graupel

may get a negative charge through riming between −20 and −10 °C. Therefore, the composite negative charge on graupel exceeds the positive charge generated by noninductive charging. The ice or snow mainly carries a positive charge below 10 km (Fig. 12g), indicating significant sedimentation of snow crystals generated between 8 and 10 km, and the positive charge carried by these falling snow crystals exceeds the negative charge transferred to snow through noninductive charging below 8 km. The enhanced noninductive charging rate by rime splintering resulted in a positive (negative) charge on graupel (snow) below 7 km (Fig. 13b and h), indicating that the positive charge on graupel gained from charge separation at this level exceeds the negative charge carried by the falling graupel. Above 7 km, the negative charge carried by graupel is weakened, probably due to the enhanced positively inductive charging (Fig. 12b and h). The ice–ice collisional breakup and sublimational breakup of ice enhance ice concentration after 01:00 (Fig. 9), but the relatively low graupel and droplet concentrations after 01:00 prevent the intensification of charge separation; this explains why collision between ice crystals has a weaker impact than rime splintering and drop shattering on cloud electrification in this case.

To better understand the different vertical distributions of charge density and charging rate, the cross-sections of the modeled graupel charge density and noninductive charging rate from the six experiments are shown in Fig. 14. The charge densities of graupel (Fig. 14a–f) are in agreement with the distribution of graupel and ice and/or snow concentrations, which reveals the importance of the ice-phase particle number concentration for cloud electrification. The cross-

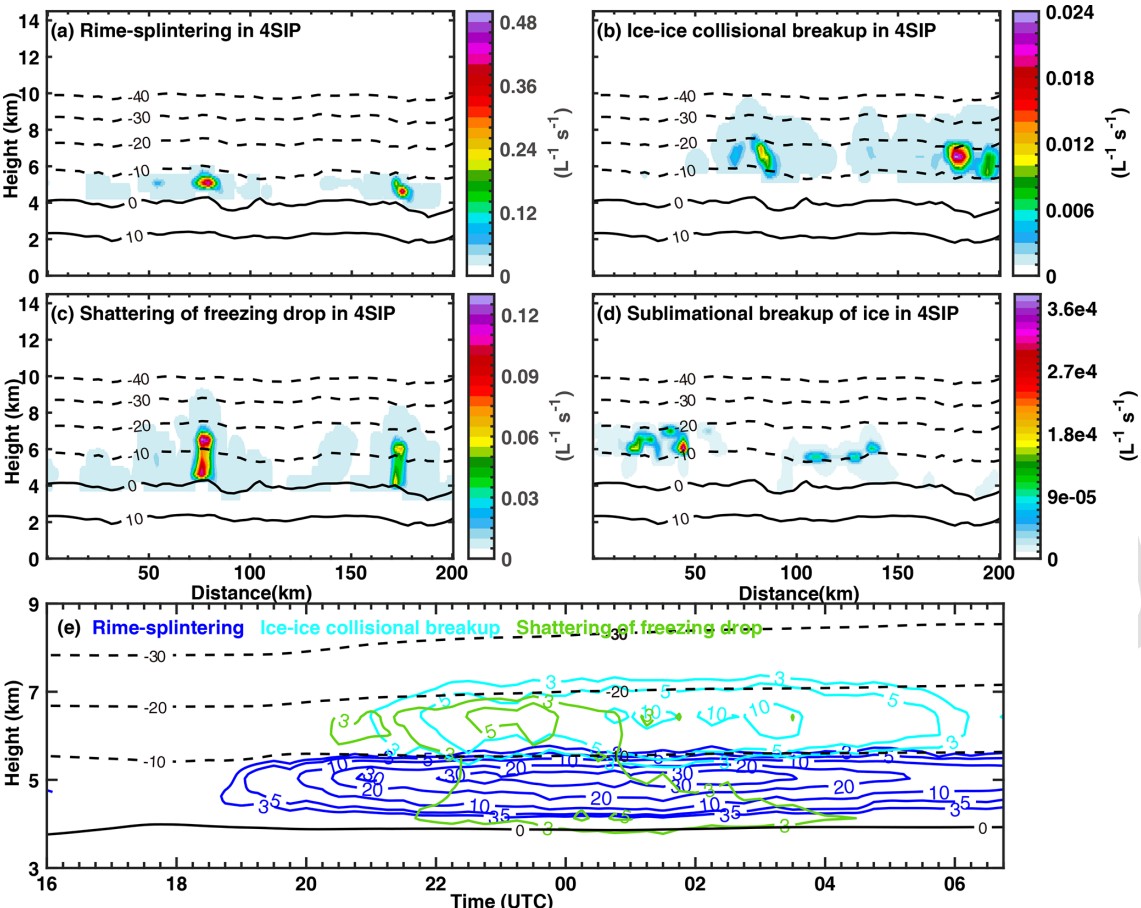

**Figure 11.** Cross-sections of the secondary ice production rates by different SIP processes resulting from the 4SIP experiment at 01:00 on 28 November. **(a)** Rime splintering, **(b)** ice–ice collisional breakup, **(c)** shattering of freezing drops, **(d)** sublimational breakup of ice, and **(e)** the time–height diagram of the mean ice production rate by different SIP processes. Contour levels are $3 \times 10^{-3}$, $5 \times 10^{-3}$, $10 \times 10^{-3}$, $20 \times 10^{-3}$, and $30 \times 10^{-3}\,\mathrm{L}^{-1}\,\mathrm{s}^{-1}$; the ice production rate of the sublimational breakup of ice is so small that it never meets the lowest contour level.

sections of the noninductive charging rate exhibit a distribution of upper negative and lower positive (Fig. 14g–l), indicating that the upper graupel particles get negative charges and that the lower graupel particles get positive charges. Since a threshold of $\mathrm{RAR} > 0.1\,\mathrm{g\,m}^{-3}\,\mathrm{s}^{-1}$ is required to trigger charge separation, charging only occurs in areas with a relatively high graupel concentration, while the fall of graupel with a negative charge is found in more areas. If the magnitude of the low-level positive charging rate is small, the average charge density would be negative; on the other hand, if the magnitude of the low-level positive charging rate is enhanced by SIP, the average low-level charge density on graupel is positive.

The above analysis is also valid when considering noninductive charging only as indicated by a sensitivity test in which inductive electrification is turned off. Figure 15 shows the graupel charge density, noninductive charging rate, and the fraction of the area with charge separation occurring in this sensitivity test. In the noSIP experiment, the graupel charge density is negative, while the noninductive charging rate has a dipole structure. The magnitude of the low-level positive charging rate is much smaller than the high-level negative charging rate. This result is the same as that shown in Figs. 12 and 13, in which both noninductive and inductive charging are considered. Therefore, it is evident that the charge density is mainly controlled by noninductive charging. Although positive charging takes place at temperatures warmer than $-20\,°\mathrm{C}$, its magnitude is small, and charging only occurs in a small fraction of the cloud area (Fig. 15e and f); thus, the average charge density on graupel is negative. With rime splintering implemented, the low-level positive charging is substantially enhanced, and the average charge density on graupel is positive at temperatures warmer than $-20\,°\mathrm{C}$.

The time–height evolution of the total charge density obtained from different simulations is shown in Fig. 16. In the experiment without any SIP (Fig. 16a), the storm has an inverted tripole structure with a positive-charge region at 7–

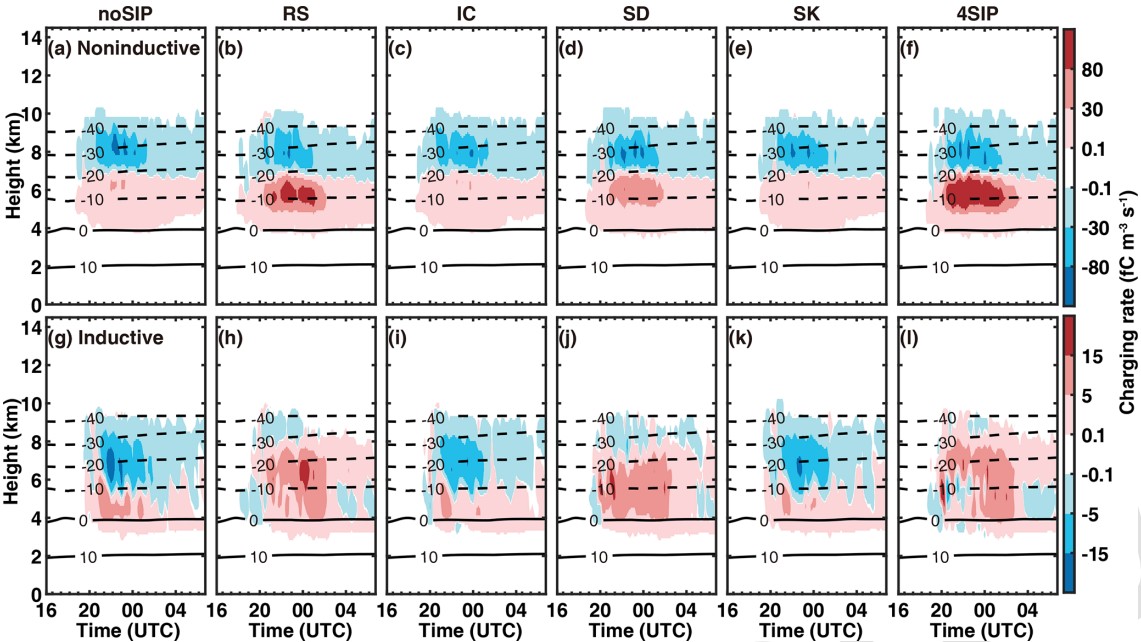

**Figure 12.** Time–height diagrams of the charging rate on graupel through noninductive (left panels) and inductive (right panels) charging from the six experiments. **(a, b)** Experiment without SIP, **(c, d)** experiment with rime splintering, **(e, f)** experiment with ice–ice collisional breakup, **(g, h)** experiment with shattering of freezing drops, **(i, j)** experiment with sublimational breakup of ice, and **(k, l)** experiment with four SIP processes. The black contours are the isotherms.

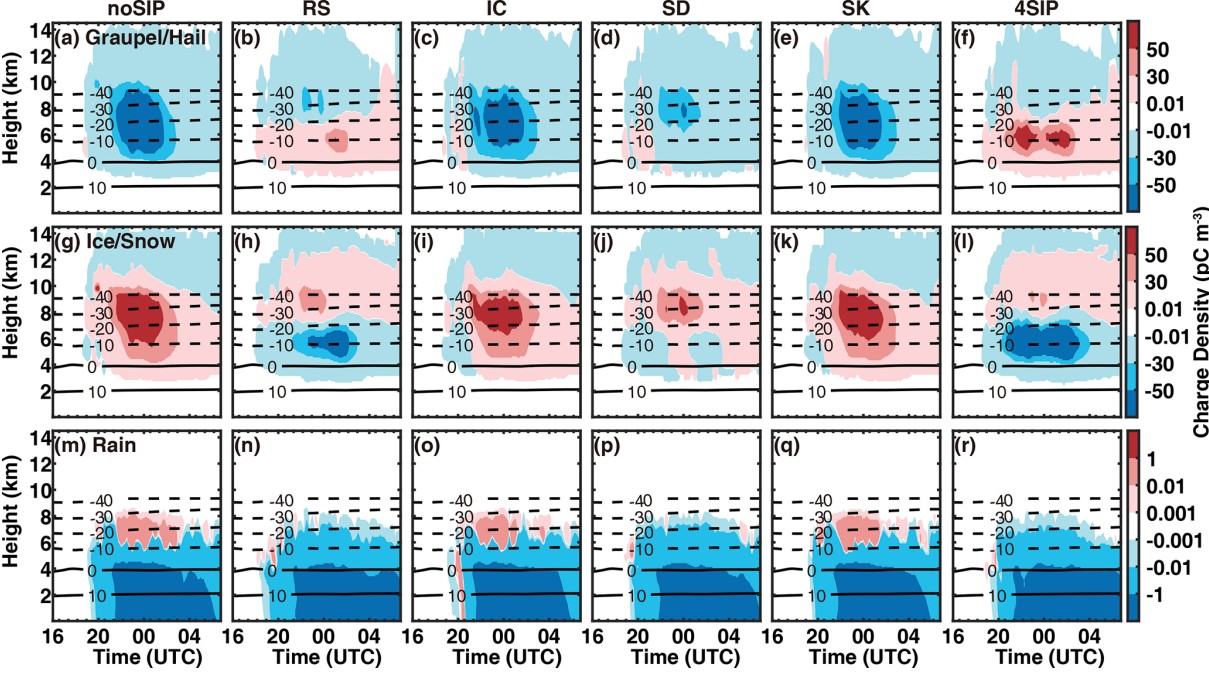

**Figure 13.** Time–height diagrams of the charge density carried by **(a–f)** graupel and/or hail, **(g–l)** ice and/or snow, and **(m–r)** rain from the six simulations. **(a, g, m)** Experiment without SIP, **(b, h, n)** experiment with rime splintering, **(c, i, o)** experiment with ice–ice collisional breakup, **(d, j, p)** experiment with shattering of freezing drops, **(e, k, q)** experiment with sublimational breakup of ice, and **(f, l, r)** experiment with four SIP processes. The black contours are the isotherms.

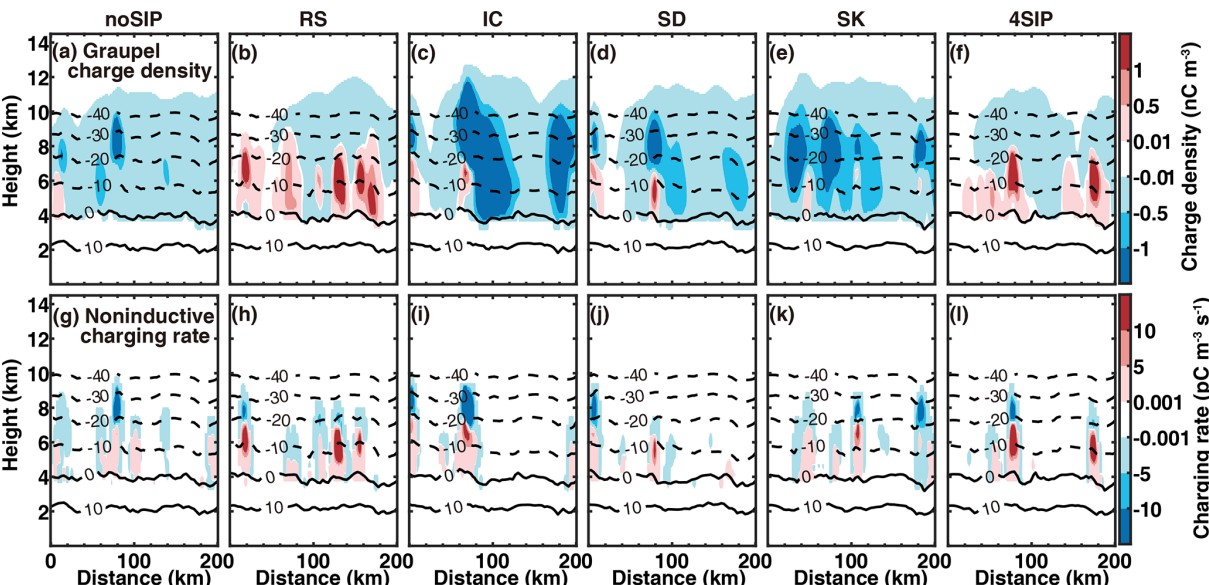

**Figure 14.** Cross-sections of the modeled **(a–f)** graupel charge density and **(g–l)** noninductive charging rate.

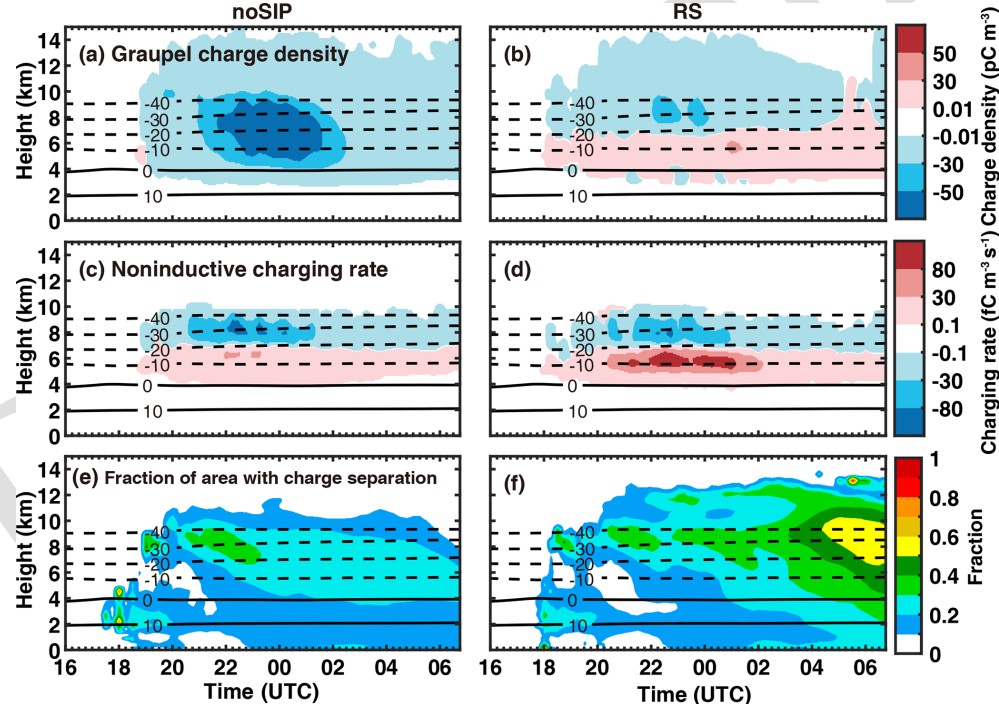

**Figure 15.** Time–height diagrams of **(a, b)** graupel charge density, **(c, d)** noninductive charging rate, and **(e, f)** fraction of area with charge separation occurring in noSIP and RS experiments with only noninductive charging used.

10 km and an upper and a lower negative-charge region. The positive-charge region weakened after 02:00 on 28 November due to the lower positive charging rate (Fig. 12a). With rime splintering implemented, the charge density changes to a dipole structure on average (Fig. 16b). The main positive charge dominated above 8 km, while the main negative charge dominated below 8 km. A weak negative-charge layer

is present at the cloud top. This indicates that the magnitude of charge carried by ice and/or snow is larger than that carried by graupel and/or hail (Fig. 13b and h). With the four SIP processes included, the charge structure is dipole as well, suggesting that the rime splintering dominates the SIP effect. In addition, it is seen that the charge reversal level shifts upwards by about 1 km and that the magnitude of the upper-

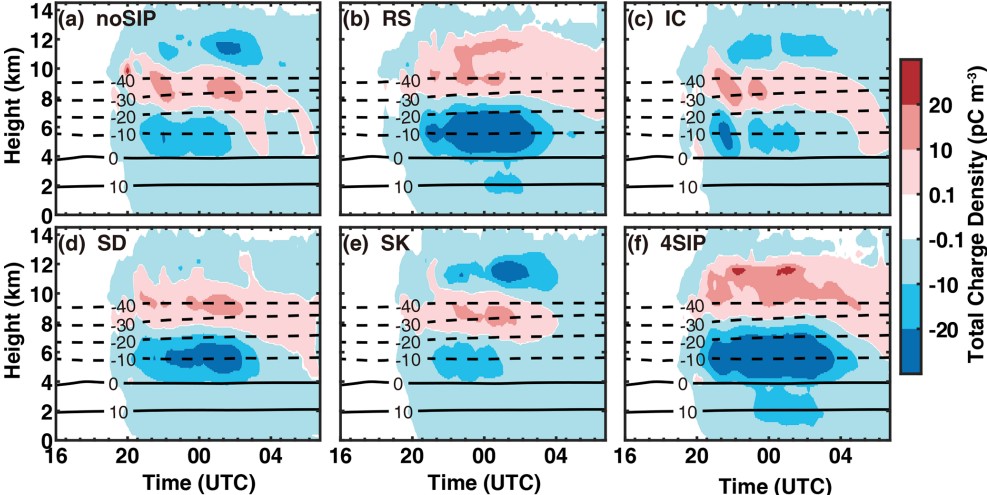

**Figure 16.** Time–height diagrams of the total charge density (colored) and temperature (contours) from the six numerical experiments. **(a)** Experiment without SIP, **(b)** experiment with rime splintering, **(c)** experiment with ice–ice collisional breakup, **(d)** experiment with shattering of freezing drops, **(e)** experiment with sublimational breakup of ice, and **(f)** experiment with four SIP processes.

level positive-charge density is greater compared to that in the RS and SD experiments due to the composite effect of the SIP processes.

The structure of the average charge density shown Fig. 16 looks fairly simple; however, the actual charge structure along a given cross-section is complicated. Figure 17 shows the cross-section of the total charge density. In general, if no SIP is considered, there is a main upper negative-charge region and a main middle positive-charge region, and a negative-charge region is observed sometimes at the bottom of the cloud. The IC and SK experiments show a similar structure to noSIP. But the charge structure could be different at different locations, suggesting complicated microphysics processes. Due to the presence of small positive-charge regions at low levels, the charge structures in RS and SD experiments vary significantly along the cross-section (Fig. 17b and d). With all the SIP processes considered, the storm obtains a different charge structure compared to that in the noSIP experiment as there is a main positive-charge region at the top and a main negative-charge region below. Small positive-charge regions are present at some locations near $-10\,°C$, but it cannot be intuitively revealed after averaging (Fig. 16f). The substantial change in the charge structure induced by SIP suggests that the charge separation in this storm is very sensitive to the ice and graupel generation (i.e., increase in ice and graupel mixing ratio and number concentration).

The importance of the increase in graupel and ice concentration can be better interpreted according to Eq. (B1), shown in Appendix B, in which we can see that the charge transfer is determined by three terms: (1) charge transferred during each collision between graupel and ice ($\delta q_{gi}$), (2) collision kernel between graupel and ice, and (3) concentration of graupel and ice. $\delta q_{gi}$ is determined by RAR, which is a function of liquid-water content (LWC) and the terminal velocity of graupel. With the addition of SIP, the LWC generally decreases (Fig. 8), and the diameters of ice particles decrease as well (Fig. 10), leading to a decrease in RAR (Fig. 18), especially in RS and SD experiments. The collision kernel between graupel and ice is determined by the terminal velocity and size of graupel and ice, which also decrease after SIP processes are implemented. The concentration of graupel ($n_g$) and ice ($n_i$) increases due to the rime splintering and shattering of freezing drops; this explains the enhanced electrification by these two SIP processes.

Changes in the structure of total charge density result in changes in the electric field by the SIP processes. Figure 19 shows the time–height diagram of the vertical electric field modeled in different experiments. It is evident that the electric field is enhanced by the SIP, especially by the rime splintering and shattering of freezing drops. The IC and SK experiments have a similar electric field to noSIP. The rime splintering and shattering of freezing drops enhance the vertical electric field after 00:00 on 28 November (Fig. 18b and d). With all implemented, the eclectic field is enhanced, especially after 00:00 on 28 November (Fig. 18f), resulting in higher lightning frequency in the entire period.

## 4 Discussion and conclusions

In this study, the impacts of different SIP processes on cloud electrification in a cold-season thunderstorm are investigated using WRF model simulations with an SBM microphysics scheme. The storm occurred in late November in southeastern China. Four SIP processes are considered in the model, including the rime splintering, the ice–ice colli-

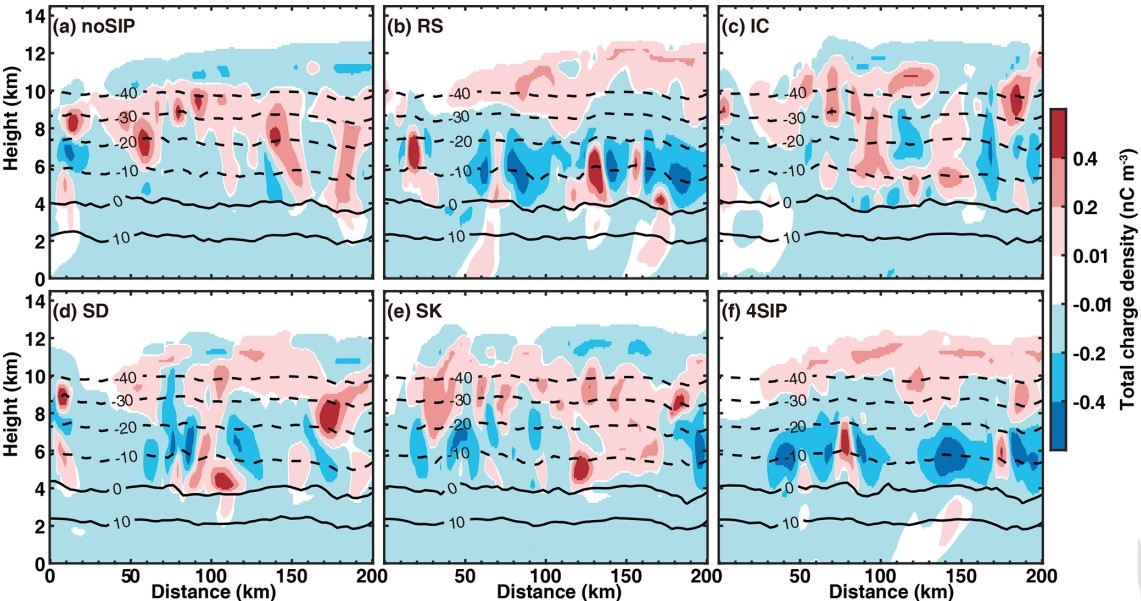

**Figure 17.** Cross-sections of the modeled total charge density from the six numerical experiments. **(a)** Experiment without SIP, **(b)** experiment with rime splintering, **(c)** experiment with ice–ice collisional breakup, **(d)** experiment with shattering of freezing drops, **(e)** experiment with sublimational breakup of ice, and **(f)** experiment with four SIP processes.

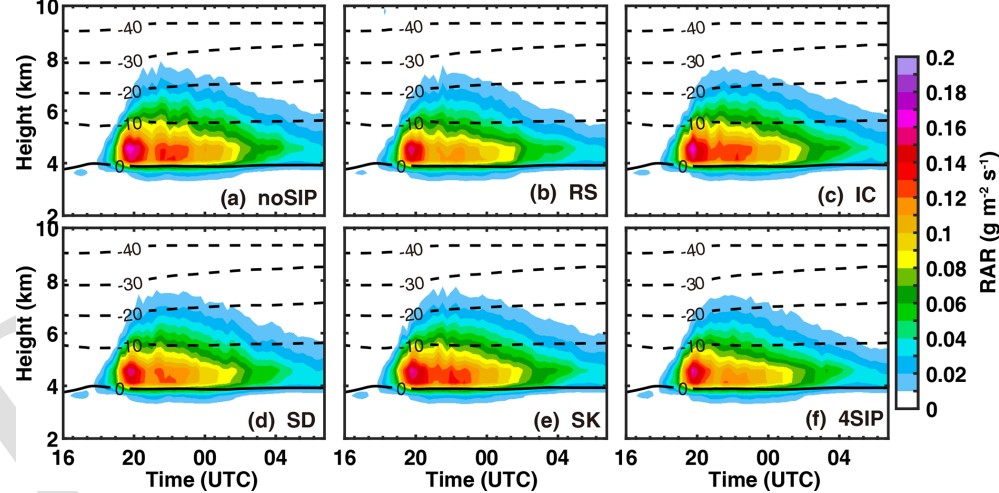

**Figure 18.** Time–height diagrams of the RAR from the six numerical experiments. **(a)** Experiment without SIP, **(b)** experiment with rime splintering, **(c)** experiment with ice–ice collisional breakup, **(d)** experiment with shattering of freezing drops, **(e)** experiment with sublimational breakup of ice, and **(f)** experiment with four SIP processes.

sional breakup, the shattering of freezing drops, and the sublimational breakup of ice. In addition, a noninductive charging parameterization and an inductive charging parameterization, as well as a bulk discharging model, are coupled with the SBM microphysics. The impacts of different SIP processes on cloud microphysics and electrification are compared using six sensitivity experiments: one control run without SIP, one with all four SIP processes, and four where a single SIP is used in each. The results contribute to filling the dearth in terms of understanding the impact of different

SIP processes on cloud electrification in cold-season thunderstorms.

Comparison between model simulations and observations suggests that the model captures well the scale and eastward propagation of the storm. The SIP has minor impacts on the macro-properties of the storm, while the intensity can be affected. If no SIP is considered, the model overestimates the composite radar reflectivity. With all implemented, the simulation result (composite radar reflectivity and CFAD) is more consistent with the observation. This is mainly because the

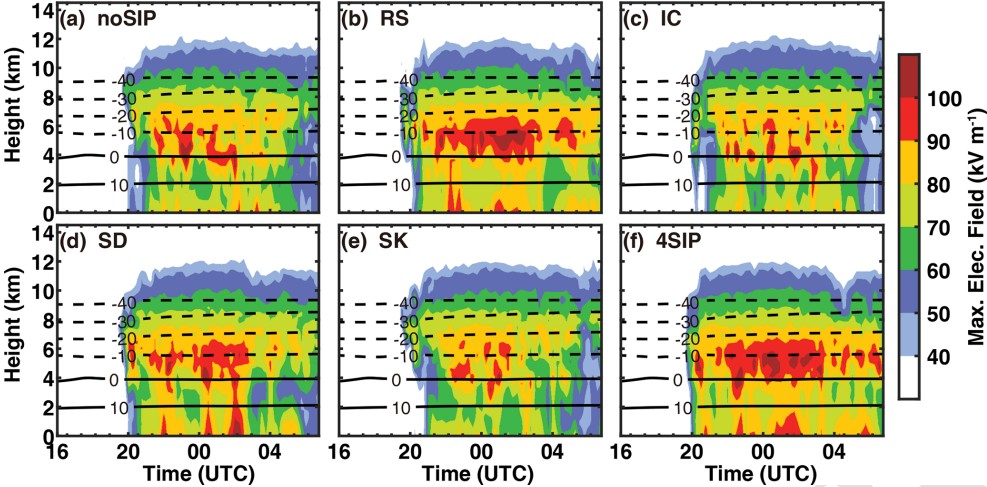

**Figure 19.** Time–height diagrams of the maximum vertical electric field from the six numerical experiments. **(a)** Experiment without SIP, **(b)** experiment with rime splintering, **(c)** experiment with ice–ice collisional breakup, **(d)** experiment with shattering of freezing drops, **(e)** experiment with sublimational breakup of ice, and **(f)** experiment with four SIP processes.

SIP processes suppress the sizes of graupel and snow, though their concentration can be enhanced. The implementation of SIP also improves the simulation of flash rates. Without any SIP, the lightning activity dissipated more rapidly. With all implemented, both the temporal variation and magnitude of the flash rate are more consistent with the observation.

Different SIP processes have different impacts on cloud microphysics electrification. The rime splintering and shattering of freezing drops are active throughout the cloud life cycle but are limited to relatively warm temperatures. The cloud glaciation below 8 km is enhanced by these two processes, leading to lower LWC at higher levels. The low-level positive charging is significantly enhanced by them due to the higher graupel and ice and/or snow concentrations. The ice–ice collisional breakup is more active in regions with higher ice and/or snow mixing ratios; its average impact on cloud electrification is minor, while it could be significant in some areas in the cloud. The sublimational breakup of snow is more active near cloud edges or in downdrafts, and its average impact on cloud electrification is weak. Among the four SIP processes, rime splintering has the greatest impact on cloud microphysics, and its ice production rate is higher than that of the others, while the impact of sublimational breakup of ice is the weakest, and its ice production rate is the lowest.

In the case presented in this paper, the noninductive charging rate has a reversal at $-20\,°C$, with negative charging on graupel above this level and positive charging below. Without SIP considered, the magnitude of the upper-level negative charging rate is larger than the positive charging rate. With rime splintering or shattering of freezing drops included, the positive charging rate is substantially enhanced. The inductive charging rate is a few times smaller than the noninductive charging rate, and the SIP can change the upper-level inductive charging on graupel from negative to positive. The

changes in the charging rate due to SIP result in substantial modification of the charge structure. The charge density carried by graupel and snow below $-20\,°C$ obtains an opposite sign after SIP is implemented in the model. The total charge density changes from an inverted tripole structure to a dipole structure (tripole structure at some locations) after four SIP processes are implemented in the model. These changes lead to an enhancement of the vertical electric field, especially in the mature stage.

Due to the scarcity of winter thunderstorms, there have been few modeling studies of them. Takahashi et al. (2019) studied the winter clouds in Hokuriku and found that lightning was generated in clouds with the following conditions: cloud top temperature less than $-14\,°C$, $-10\,°C$ isotherm is higher than $1.2\,km$, space charge greater than $2$–$3\,pC\,L^{-1}$, ice crystal concentration greater than $500\,m^{-3}$, and graupel concentration greater than $20\,m^{-3}$. According to the analysis above, the thundercloud studied in this paper satisfies all these characteristics. Takahashi et al. (2017) pointed out that winter thunderstorm clouds have lower LWC and lower cloud tops than summertime convections. In our simulation, the modeled LWC is typically lower than $1\,g\,m^{-3}$, which is lower than that reported in summer convective clouds (e.g., Yang et al., 2016; Phillips and Patade, 2022). The lower LWC in wintertime convection indicates weaker riming and thus a lower RAR, which potentially leads to a higher possibility of an inverted charge structure of thunderstorms (Wang et al., 2021). In many previous studies of summertime thunderstorms that occurred at a similar latitude (e.g., Caicedo et al., 2018; Shi et al., 2015), the main charging region is typically at $5$–$11\,km$ a.m.s.l., and the freezing level is at about $5\,km$ a.m.s.l., both of which are about $1\,km$ higher than the cold-season storm shown in this paper.

Some studies suggest that charge separation in thunderstorms is sensitive to the parameterization of electrification (Altaratz et al., 2005; Fierro et al., 2013; Xu et al., 2019). Here, we highlight that the cold-season cloud electrification is also sensitive to the SIP. However, the results shown here only reveal the relative importance of four SIP mechanisms in a single case. In other cases, the SIP processes may have different impacts on the charge structure. For example, Phillips and Patade (2022) suggested that, in summertime thunderstorms with a high cloud base, the ice–ice collisional breakup has stronger impacts than the other SIP mechanisms, which is different from the result shown in this paper. Huang et al. (2022) analyzed the relative contribution of three SIP processes to ice generation using model simulations; they compared the modeled microphysics to airborne observations, and the results showed that shattering of freezing droplets dominates ice particle production at temperatures between $-15$ and $0\,^{\circ}\mathrm{C}$ during the developing stage of convection, and ice–ice collisional breakup dominates at temperatures shown during the later stage of convection. Studies that investigate the impacts of different SIPs on cloud electrification are still limited. It will be interesting to see how changes in different environmental conditions (such as wind shear, cloud base height, and aerosol concentrations) in different cases would influence the role of different SIPs. Based on the results in this study, it is suggested that sufficient graupel is important for SIP processes to enhance cloud electrification.

Future work includes more studies of different cases and improvement of the parameterizations. Currently, there are still some assumptions used in the parameterizations – for instance, the rimed fraction of ice crystals, which influences the efficiency of the ice–ice collision (Karalis et al., 2022; Sotiropoulou et al., 2021), is assumed to be 0.2 in this study. Laboratory and field measurements would be helpful to determine these parameters. Some other ice processes that are not considered in the model may also influence cloud electrification, such as ice fragmentation due to thermal shock (Korolev et al., 2019) and pre-activation of ice nucleating particles (Jing et al., 2022). It is worth investigating the impacts of these mechanisms using model simulations once there are sufficient measurements to support the development of parameterizations in the future.

## Appendix A

Based on laboratory experiments, Hallett and Mossop (1974) showed that one ice splinter can be generated during the riming process for every 200 droplets collected by a graupel. The ice splinter production rate of rime splintering $N_{\mathrm{RS}}$ is

$$N_{\mathrm{RS}} = 3.5 \times 10^5 \cdot \left(\frac{\partial m_{\mathrm{g}}}{\partial t}\right) \cdot R_{\mathrm{rim}}(T), \qquad (A1)$$

$$R_{\mathrm{rim}}(T) = \begin{cases} 0, & T \geq 270.16\,\mathrm{K} \\ (T-268.16)/2, & 268.16\,\mathrm{K} \leq T < 270.17\,\mathrm{K} \\ (T-268.16)/3, & 265.16\,\mathrm{K} \leq T < 268.16\,\mathrm{K} \\ 0, & T < 265.16\,\mathrm{K} \end{cases}, \qquad (A2)$$

where $\frac{\partial m_{\mathrm{g}}}{\partial t}$ indicates the riming rate, and $T$ is the temperature.

The parameterization of ice–ice collisional breakup is developed by Phillips et al. (2017). The number of ice fragments produced during ice–ice collision is

$$N_{\mathrm{IC}} = \alpha A(M) \left\{ 1 - \exp\left[ -\left(\frac{C(M)K_0}{\alpha A(M)}\right)^{\gamma} \right] \right\}, \qquad (A3)$$

where $A(M)$ is the number density of breakable asperities on the ice particle and is related to the rimed fraction and the size of smaller ice particle; $C(M)$ is the asperity–fragility coefficient that is set as $3.86 \times 10^4$ according to the cloud chamber experiment of natural ice particles (Gautam, 2022); $K_0$ is the initial value of collision kinetic energy; and $\gamma$ and $\alpha$ are the shape parameter and the equivalent spherical surface area of smaller particles, respectively. $\gamma = 0.5 - 0.25\Psi$, where $\Psi$ denotes the rimed fraction, which is assumed to be 0.2 in this study. The tiny fragments are treated as the ice particles belonging to the first bin of the fast-SBM model. In the WRF SBM, the collision efficiency between ice crystals is obtained based on Bohm's theory (Bohm, 1992a, b) and the superposition method in Khain et al. (2001). The coalescence efficiency is parameterized based on Khain and Sednev (1995) as a function of vapor pressure and temperature.

The parameterization of the shattering of freezing drops was developed by Phillips et al. (2018) based on laboratory experiments. If making contact with a smaller ice particle, a supercooled drop may break up and produce both big and tiny ice fragments; thus, the number of the ice fragments can be expressed using

$$N_{\mathrm{SD\_1}} = N_{\mathrm{T}} + N_{\mathrm{B}}, \qquad (A4)$$

$$N_{\mathrm{SD\_1}} = F(D)\Omega(T)\left[ \frac{\xi_{\mathrm{T}}\eta_{\mathrm{T}}^2}{(T-T_{\mathrm{T},0})^2 + \eta_{\mathrm{t}}^2} + \beta T \right], \qquad (A5)$$

$$N_{\mathrm{B}} = \min\left\{ F(D)\Omega(T)\left[ \frac{\xi_{\mathrm{B}}\eta_{\mathrm{B}}^2}{(T-T_{\mathrm{B},0})^2 + \eta_{\mathrm{B}}^2} \right], N_{\mathrm{T}} \right\}, \qquad (A6)$$

where $N_{\mathrm{T}}$ and $N_{\mathrm{B}}$ are the number of tiny and big ice fragments generated by a shattered drop. $F(D)$ and $\Omega(T)$ are the interpolating functions for the onset of drop shattering. $\xi_{\mathrm{T}}$, $\xi_{\mathrm{B}}$, $\eta_{\mathrm{T}}$, $\eta_{\mathrm{B}}$, $T_{\mathrm{T},0}$, $T_{\mathrm{B},0}$, and $\beta$ are parameters determined based on datasets from previous laboratory experiments, which can be found in Phillips et al. (2018). The tiny fragments are treated as the ice particle belonging to the first bin of the fast-SBM model, with a diameter of $4\,\mu\mathrm{m}$ (Khain et al., 2004). The mass of big ice fragments is $m_{\mathrm{B}} = 0.4 m_{\mathrm{drop}}$.

In addition, a drop may also break if coming into contact with a more massive ice particle. The number of ice fragments produced in this process is

$$N_{SD\_2} = 3\Phi \times [1 - f(T)] \times \max\left\{\left(\frac{k_0}{S_e} - DE_{crit}\right), 0\right\}, \quad \text{(A7)}$$

$$f(T) = \frac{-C_w T}{L_f}, \quad \text{(A8)}$$

$$S_e = \gamma_{liq}\pi D^2, \quad \text{(A9)}$$

where $\gamma_{liq}$ is the surface tension of a liquid drop, $k_0$ is the initial kinetic energy of the two colliding particles, and $f(T)$ is the frozen fraction. $C_w$ and $L_f$ are the specific heat capacity of water and the specific latent heat of freezing, respectively. $DE_{crit} = 0.2$, and $\Phi$ is 0.3 according to James et al. (2021). All ice fragments are assumed to be tiny in this mode. The tiny ice fragments are added to the first bin of ice size distribution.

The parameterization of the sublimational breakup of ice is proposed by Deshmukh et al. (2022). The number of ice splinters produced during sublimation is dependent on the relative humidity on the ice and the preliminary size of the mother ice particles. The formulation is used for dendritic crystals and heavily rimed particles (e.g., graupel). The rate of ice splinters produced by dendritic crystals is

$$\frac{dN}{dt} \approx \beta d^\gamma d(100 - RH_i)f_\upsilon \Xi \nu, \quad \text{(A10)}$$

$$\nu(RH_i, d) = \Delta_0^1[RH_i RH_{i0}(d), RH_{i0}(d) + \Delta RH_i], \quad \text{(A11)}$$

$$RH_{i0} = 72\lambda + 94(1 - \lambda), \quad \text{(A12)}$$

where $d$ refers to the diameter of parent ice particles, $RH_i$ represents the relative humidity over ice, $f_\upsilon$ denotes the ventilation coefficient for vapor diffusion, $\Xi$ is the emission factor, $\nu$ is the onset transition factor for dendrites, and $\lambda$ is the size-dependent fraction. $\beta$ and $\lambda$ are empirical parameters. $\Delta RH_i = 6\%$.

$$\frac{dN}{dt} \approx \frac{\rho_{D0}}{\rho_r}\beta d^\gamma d(100 - RH_i)f_\upsilon \Xi \nu^* \quad \text{(A13)}$$

$$\nu^*(RH_i, d) = \Delta_0^1[RH_i RH_{i0}(d), RH_{i0}(d) + (1 + 2\lambda)\Delta RH_i] \quad \text{(A14)}$$

In the above, $\rho_r$ denotes the density of a rimed particle. $\rho_{D0}$ is the density observed by Dong et al. (1994). $\rho_{D0} = 300\,\text{kg m}^{-3}$. $\nu^*$ is the onset transition factor for graupel, and the mass of ice fragments is $m_f = \chi m_{ice}$.

## Appendix B

The noninductive charging produced during the collision between graupel and ice crystal is expressed as

$$\frac{\partial \rho_{gi}}{\partial t} = \iint_0^\infty \frac{\pi}{4}\beta\delta q_{gi}(1 - E_{gi})|V_g - V_i|$$
$$\times (D_g + D_i)^2 n_g n_i dD_g dD_i \quad , \quad \text{(B1)}$$

$$\beta = \begin{cases} 1, & T > -30° \\ 1 - \left[\frac{T+30}{13}\right]^2, & 43° < T < -30° \\ 0, & T < -43° \end{cases} \quad \text{(B2)}$$

where $T$ is temperature. $E_{gi}$ is the collection efficiency between graupel and ice. $V$, $D$, and $n$ are the terminal velocity, diameter, and number concentration, with subscripts $g$ and $i$ indicating graupel and ice crystals. The charge transferred per rebounding collision ($\delta q_{xy}$) is a function of rime accretion rate (RAR) and critical RAR ($RAR_C$) (Saunders and Peck, 1998):

$$\delta q_{xy} = B d^a V^b \delta q_\pm, \quad \text{(B3)}$$

where $B$, $a$, and $b$ are parameters determined based on laboratory studies. For positive charging of graupel, (RAR > $RAR_C$),

$$\delta q_+ = 6.74(RAR - RAR_C). \quad \text{(B4)}$$

For negative charging ($0.1\,\text{g m}^{-2}\,\text{s}^{-1} < RAR < RAR_C$),

$$\delta q_- = 3.9(RAR_C - 0.1)$$
$$\times \left\{4\left[\frac{RAR - (RAR_C + 0.1)/2}{RAR_C - 0.1}\right]^2 - 1\right\}, \quad \text{(B5)}$$

$$RAR_C = \begin{cases} s(T), & T > -23.7° \\ k(T), & -23.7° > T > -40° \\ 0, & T \le -40°, \end{cases} \quad \text{(B6)}$$

$$s(T) = 1.0 + 7.9262 \times 10^{-2}T + 4.4847 \times 10^{-2}T^2$$
$$+ 7.4754 \times 10^{-3}T^3 + 5.4686 \times 10^{-4}T^4$$
$$+ 1.6737 \times 10^{-5}T^5 + 1.7613 \times 10^{-7}T^6, \quad \text{(B7)}$$

$$k(T) = 3.4[1.0 - \left(\frac{|T + 23.7|}{-23.7 + 40.0}\right)^3]. \quad \text{(B8)}$$

According to Mansell et al. (2005), the inductive charging rate is parameterized as

$$\frac{\partial \rho_g}{\partial t} = \left(\frac{\pi^3}{8}\right)\left(\frac{6.0\overline{V}_g}{\Gamma(4.5)}\right)E_{gc}E_r n_c n_{0g} D_c^2$$
$$\times [\pi \Gamma(3.5)\epsilon\langle\cos\theta\rangle E_z D_g^2 - \Gamma(1.5)\frac{\rho_g}{3n_g}], \quad \text{(B9)}$$

where $E_{gc}$ is the collision efficiency between graupel and droplet. $E_r$ is the rebound probability. $n_c$ is the number concentration of cloud droplets. $n_{0g}$ is the intercept of the graupel size distribution. $\theta$ is the rebounding collision angle. $\epsilon$ is the permittivity of air. $E_z$ is the vertical electric field, and $\rho_g$ is the charge density carried by graupel.

The discharge model used in this paper is a bulk discharge scheme suggested by Fierro et al. (2013), in which flash occurs once the electric field exceeds a threshold. The electric field ($E$) can be computed by solving the Poisson equation:

$$\nabla^2 \varnothing = -\frac{\rho_{tot}}{\epsilon}, \tag{B10}$$

$$E = -\nabla\varnothing, \tag{B11}$$

where $\rho_{tot}$ is the net charge density.

**Data availability.** The WRF model is available on https://www2.mmm.ucar.edu/wrf/users/download/get_source.html (NCAR MMM, 2023). The reanalysis data used to drive the WRF model, the observed radar reflectivity, sounding data, and lightning data are available at https://doi.org/10.5281/zenodo.8371845 (Yang, 2023).

**Author contributions.** SH and JY implemented the parameterizations of SIP and electrification in WRF and designed the numerical experiments. SH, JY, and YL performed the analysis and prepared the paper. TY, QZ, and YD contributed to the model evaluation. TY and QZ provided input on the method and analysis. All the authors provided significant feedback on the paper.

**Competing interests.** The contact author has declared that none of the authors has any competing interests.

**Disclaimer.** Publisher's note: Copernicus Publications remains neutral with regard to jurisdictional claims made in the text, published maps, institutional affiliations, or any other geographical representation in this paper. While Copernicus Publications makes every effort to include appropriate place names, the final responsibility lies with the authors.

**Acknowledgements.** We acknowledge the High Performance Computing Center of Nanjing University of Information Science and Technology for their support of this work. We appreciate the editor and reviewers for their insightful comments and suggestions.

**Financial support.** This research has been supported by the NSFC-CMA Joint Research (grant no. U2342222); the National Natural Science Foundation of China (grant no. 41905124); the Natural Science Foundation of Jiangsu Province, China (grant no. BK20190778); and the CMA Key Innovation Team Support Project (grant no. CMA2022ZD10).

**Review statement.** This paper was edited by Corinna Hoose and reviewed by three anonymous referees.

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

## Remarks from the typesetter