# Peer review of "Impact of ice multiplication on the cloud electrification of a coldseason thunderstorm: a numerical case study"

_EGUsphere, 2023_

## Referee Comment (RC1)

Review of the paper "**Impact of Ice Multiplication on the Cloud Electrification of a cold-season thunderstorm: a numerical case study**" by Jing Yang et al.

**General Comments:**

Yang et al studied the role of three secondary ice production (SIP) mechanisms on cloud electrification in a simulated thunderstorm that was developed during the cold season. They implemented three major SIP mechanisms in the WRF model with fast SBM microphysics along with inductive and non-inductive charging mechanisms. Overall, the effect of SIP mechanisms on electrification is an important topic for the scientific community. However, in the current format, the paper needs major revision. Authors need to improve most of the sections including model validation, Analysis, implementation of SIPs, etc. I have enlisted specific and minor comments below.

**Specific comments:**

1) In the present study model validation is only based on spatial distribution radar reflectivity and temporal evolution flash rates. Since the study considers 3 major SIP processes, to what extent does the model agree with the observed number of concentrations of ice particles? How well does the model simulate the liquid water mass/content and vertical velocities? All these microphysical properties are of great importance for lightning. Comparison of some of these microphysical properties with the observation will be helpful for readers to understand the accuracy of the model. It will be good to compare the vertical distribution of radar reflectivity from the model with observations. If available, surface precipitation can be compared to show the robustness of model simulations.

2) Even with radar reflectivity plots, contour levels are different in observations and model, which makes it difficult to compare. To what extent does the simulated radar reflectivity is

in agreement with observations when all SIP processes are active? It will be good to present some statistical analysis.

3)  Based on my knowledge, in most of the previous studies, ice-ice collision is a major SIP mechanism in deep convective clouds when compared with rime-splintering and drop shattering (e.g. Phillips and Patade 2022). This is because rime-splintering and drop shattering are active over a limited range of temperatures. The authors need to mention the reasons behind less active ice ice collision in the simulated case. What are the factors that resulted in high secondary production by rime-splintering and drop shattering when compared with ice-ice collisions? What are the major differences in the microphysical processes of wintertime thunderstorms and summertime thunderstorms? There should be some discussion on the relative role of SIP in modulating ice number concentration and hence cloud electrification.

4)  It is important to show the rates of three SIP processes implemented in the model. Or at least the concentration of ice resulting from each SIP mechanism in 3SIP simulations can be shown. Time height evolution of ice particle number concentration from each of the SIP mechanisms will help to understand their relative importance in altering total ice number concentration. Authors have shown time height evolution of mass mixing ratios, however, changes in ice number concentration are very important as far as the role of SIP is concerned.

5) In Figure 8, temporal variation of ice/snow showed that there is not much effect of individual SIP process on ice/snow concentration, however when all SIPs were considered the concentration was boosted. What are the physical mechanisms behind it? I expect a significant increase in ice/snow concentration as a result of SIP in the simulations where a single SIP is considered if that mechanism is important.

6) A few details of the implementation of SIP in the model are needed. What was the diameter of the tiny fragments that resulted from mode 1 in drop shattering?  What kind of collisions were considered for collisional breakup mechanisms?   In which category the resulting fragments were added?

7) There is no information about the radar data e.g. which radar was used, what are the data corrections etc. Similarly, there is not much information available about lightning data.

8) Line 378: if ice ice collision was less active what are the reasons behind the enhancement in the flash rate?

9) What are the mechanisms behind the improvement in the temporal distribution of flash rate in 3SIP simulations?

10) Authors should check the manuscript carefully for grammar and language corrections. In many places, articles are missing or not used properly.

**Technical corrections/Minor comments:**

1.  What was the cloud base height and temperature of simulated clouds?

2.  Figure 1 captions: The time mentioned in the caption does not match that mentioned in the plots. Also, plots 1a and 1b are supposed to be 500 mb geopotential height, isotherms and wind barbs, but on plot b the mentioned height is 850 hpa. The same mistake is with plots c and d.

3.  Figure 7: What are the averaging conditions for incloud points shown in time height plots?

4.  Figure 7 captions:  The names of sensitivity studies mentioned in the captions "SBM-0SIP Simulation; SBM-1SIP SimulationSBM-2SIP Simulation; SBM-3SIP" do not match the names on the plot. Please correct it according to the sensitivity tests mentioned in the text earlier.

5.  Line 11: in a thunderstorm **that** occurred ;

6.  Line 11: **are** investigated …

7.  Line 40: correct lighting to lightning

8.  Line 55: Phillips et al. 2020

9.  Line 67: studies **that** highlighted ….

10. Line 95: warm **mois**t ..

11. Line 111: Fig.3a not 2a

12. Line 113: Fig 3c not 2c

13. Line 124: Figure 4 not 3

14. Line 124: **a** two-way nested

15. Line 127: spin-up

16. Line 140: Incomplete sentence

17. Line 140: at **temperatures** colder than

18. Line 144: it can also **be** active …

19. Line 204: change "With all the three SIP processes implement" to "With all implemented"

20. Line 217: units should be g kg-1 and not g ks-1

21. Line 232: there **are** …

22. Line 235: graupel mixing **ratio** …

23. Line 245: correct quicky to quickly

24. Line 281: resul**t**s in changes in the

25. Line 318: delete the before that

26. Line 329: cross-section

27. Line 380: implemented

28. Line 407: replace continued by continue

29. Line 408: change  falling to it falls

30. Line 414: change on to in

31. Line 469: Define RAR and RAR$_c$

---

## Referee Comment (RC3)

**Review for:** *Impact of ice multiplication on the cloud electrification of a cold-season thunderstorm: a numerical case study*

**General comments:**

Yang et al. investigate the role of three secondary ice production (SIP) processes in precipitation intensity, cloud electrification and discharge processes, within the context of a wintertime thunderstorm. The analysis relies on mesoscale simulations conducted using the Weather Research and Forecasting (WRF) model, coupled with a fast spectral bin microphysics (SBM) scheme. The employed SBM scheme was refined through the incorporation of state-of-the-art ice multiplication formulations, complemented by the integration of noninductive and inductive charging parameterizations.

This study contributes significantly to clarifying the complex interactions between ice microphysics – particularly the poorly constrained SIP processes – and cloud electrification. Despite its importance, the manuscript requires substantial revisions across the methodology, model evaluation and results sections, aimed at improving readability and enhancing the robustness of certain findings. It is recommended that the following aspects be revisited before publication:

**Specific comments:**

- In Section 2.2, I would recommend to explain the rationale behind selecting the specific microphysics scheme, specifying the ice and liquid hydrometeor species considered in the model, and providing information on whether this scheme has undergone evaluation in similar studies in the past.
- Regarding the implementation of the ice-ice collisional break-up (IC) and the shattering of freezing drops (SD), it is important to provide a more detailed description – especially if this is the first attempt to incorporate these parameterizations into the SBM scheme:
  - The physically-based parameterization of Phillips et al. (2017) explicitly considers the effect of ice habit, ice type and rimed fractions of the particles undergoing fragmentation. These parameters are not always described in models, and therefore certain assumptions have to be made. Please describe how these parameters are treated in the model and whether the scheme predicts the rimed mass fraction of colliding ice particles or if a constant value is prescribed. Given the demonstrated impact of the rimed fraction on the efficiency of the IC mechanism (e.g., Karalis et al., 2022; Sotiropoulou et al., 2021), you may consider assessing the sensitivity of your results to this parameter. Additionally, further clarification is needed regarding the

collection efficiencies of ice particles and whether all collisions between ice particles can lead to fragmentation and the generation of SIP particles.

- Please provide more details about the collisions considered in 'mode 1' of the Phillips et al. (2018) parameterization. Were collisions with ice nucleating particles (INPs) other than small ice particles taken into account? A brief description of the primary ice production mechanisms encompassed within the scheme would also be useful.

- To improve readability, please consider incorporating a dedicated paragraph (for example in Section 2) that outlines the various measurements utilized in this study, discussing any uncertainties and/or any post-processing applied to them. This applies to the radar observations (Figure 3), sounding data (Figure2) as well as the observed flash rates (Figure 6). Consider moving the information about the lightning observational dataset from the "Results" section (Lines 194-197) to the corresponding data paragraph.

- Please explain how the modeled composite reflectivity (shown in Figure 5) is derived. Which parameters (e.g., mass and concentration of ice and liquid hydrometeors) have the most influence on simulated reflectivity? In this way, the reader can better understand the changes caused when SIP is accounted for and you can better support your statement in Lines 251-252 "…the decrease in the sizes of these solid particles is **probably** the main reason of the weaker composite radar reflectivity in the 3SIP experiment".

- For improved visual comparison between model simulations (Figure 5) and radar observations (Figure 3), you may consider including all relevant subplots into a single figure. Also, ensure consistency in colorbar limits (dBZ) across visualizations.

- **Lines 176-178**: Here the reader is already wondering why activating SIP in the model leads to reduced modeled reflectivity. You could mention that this aspect will be elaborated upon in Section 3.2.

- **Lines 180-181**: "…the simulation with all the three SIP processes has the best performance comparing to the observation (Figs. 3b and 5j) ". The robustness of this statement can be enhanced by including additional statistics to complement the visual comparison.

- **Line 212**: consider using a more suitable transition sentence, especially since the charge structure will not be discussed in Section 3.2.

- **Line 218**: Please clarify the meaning of "strong correlation" in this sentence.

- With the model you have access to all production rates of important microphysical processes, like riming, aggregation, sedimentation, or the melting of graupel particles or snowflakes that could be used to support your statements throughout the text, such as Lines 218, 222, 259, 285, and 287.

- **Line 224**: Are you referring to the 'riming of cloud droplets and raindrops' rather than the 'rime-splintering process' here? Indeed, liquid hydrometeors that rime onto

graupel would typically increase its mass. However, if RS is activated, part of this rimed mass would then be transferred from the graupel to the smaller cloud ice particles.

- **Line 229-230**: Any idea why the enhancement of graupel/hail and ice/snow is followed by an increase in the cloud liquid water content (rain + cloud mass mixing ratios)? I would expect the opposite behavior, because of the Wegener–Bergeron–Findeisen (WBF) process.

- **Figure 7**: I would suggest superimposing the isotherms in this plot for better visualization of the RS temperature zone, melting layer, and temperatures where IC and SD are efficient.

- **Figure 8**: Please explain how averaged concentrations were calculated. Did you consider only in-cloud conditions? Instead of having separate plots for the number concentrations and sizes, it might be worth plotting the particle size distributions (PSDs) (i.e., d(N)/d(logD)). In this way, the reader would more easily identify both the ice enhancement caused when SIP is considered in the simulations, and the shift of the PSDs towards smaller sizes, which is crucial for capturing the correct radar reflectivity values.

- The discussion of Figure 8 in the last paragraph of Section 3.2, should be more quantitative. You mention that SIP processes can "**slightly** enhance" or "**slightly** decrease" the ice-particle or liquid-particle concentrations, respectively. Please try to quantify the ice enhancement caused when SIP is included in the model compared to the noSIP sensitivity simulation. This is an important information if you want to convince the reader of the importance of incorporating SIP processes in the model.

- In Section 3.2 or the "Discussion and Conclusions" section, consider including a discussion on the relative contribution of each SIP mechanism and a comparison of your findings with similar convective case studies from the literature.

- **Line 383-384**: The transition sentence does not have a clear connection with the rest of the paragraph.

- **Line 421**: You may want to refer to the new empirical parameterization for the sublimational break-up mechanism developed in Deshmukh et al. (2022). This mechanism has been found to be the second most dominant SIP mechanism in fast convective downdrafts (Waman et al., 2022).

**Technical corrections:**

- **Line 124**: I would suggest "**grid spacing**" instead of "resolution"
- **Line 169**: I would suggest "Model **evaluation**" instead of "Model validation"
- **Line 156**: "**correct representation**" (not representative)
- **Line 228**: "shown later", consider indicating the section where the subsequent discussion will take place. Similarly, for Line 233.
- **Line 233**: "reduced **by SIP**" (not by this SIP)

- **Line 257**: Section **3.3** (not 3.2)
- **Line 361**: Section **4 Discussion and Conclusions** (not 5)
- **Line 371**: **suggests** (not suggest)
- Please double check the reference provided for Mansell et al. (2010)

**References**

Deshmukh, A., Phillips, V. T. J., Bansemer, A., Patade, S. and Waman, D.: New Empirical Formulation for the Sublimational Breakup of Graupel and Dendritic Snow, J. Atmos. Sci., 79(1), 317–336, doi:10.1175/JAS-D-20-0275.1, 2022.

Karalis, M., Sotiropoulou, G., Abel, S. J., Bossioli, E., Georgakaki, P., Methymaki, G., Nenes, A. and Tombrou, M.: Effects of secondary ice processes on a stratocumulus to cumulus transition during a cold-air outbreak, Atmos. Res., 277, doi:10.1016/j.atmosres.2022.106302, 2022.

Mansell, E. R., Ziegler, C. L. and Bruning, E. C.: Simulated electrification of a small thunderstorm with two-moment bulk microphysics, J. Atmos. Sci., 67(1), 171–194, doi:10.1175/2009JAS2965.1, 2010.

Phillips, V. T. J., Yano, J. I. and Khain, A.: Ice multiplication by breakup in ice-ice collisions. Part I: Theoretical formulation, J. Atmos. Sci., 74(6), 1705–1719, doi:10.1175/JAS-D-16-0224.1, 2017.

Phillips, V. T. J., Patade, S., Gutierrez, J. and Bansemer, A.: Secondary ice production by fragmentation of freezing drops: Formulation and theory, J. Atmos. Sci., 75(9), 3031–3070, doi:10.1175/JAS-D-17-0190.1, 2018.

Sotiropoulou, G., Vignon, E., Young, G., Morrison, H., O'Shea, S. J., Lachlan-Cope, T., Berne, A. and Nenes, A.: Secondary ice production in summer clouds over the Antarctic coast: An underappreciated process in atmospheric models, Atmos. Chem. Phys., 21(2), 755–771, doi:10.5194/acp-21-755-2021, 2021.

Waman, D., Patade, S., Jadav, A., Deshmukh, A., Gupta, A. K., Phillips, V. T. J., Bansemer, A. and Demott, P. J.: Dependencies of Four Mechanisms of Secondary Ice Production on Cloud-Top Temperature in a Continental Convective Storm, J. Atmos. Sci., 79(12), 3375–3404, doi:10.1175/JAS-D-21-0278.1, 2022.

---

## Author Response (AR1)

Reviewers' comments are in black, and responses are in blue.

**Reviewer 1:**

**General Comments:**

Yang et al studied the role of three secondary ice production (SIP) mechanisms on cloud electrification in a simulated thunderstorm that was developed during the cold season. They implemented three major SIP mechanisms in the WRF model with fast SBM microphysics along with inductive and non-inductive charging mechanisms. Overall, the effect of SIP mechanisms on electrification is an important topic for the scientific community. However, in the current format, the paper needs major revision. Authors need to improve most of the sections including model validation, Analysis, implementation of SIPs, etc. I have enlisted specific and minor comments below.

Reply: We appreciate your insightful comments. The paper has been revised accordingly and has been improved a lot. Please see our responses below.

**Specific comments:**

1. In the present study model validation is only based on spatial distribution radar reflectivity and temporal evolution flash rates. Since the study considers 3 major SIP processes, to what extent does the model agree with the observed number of concentrations of ice particles? How well does the model simulate the liquid water mass/content and vertical velocities? All these microphysical properties are of great importance for lightning. Comparison of some of these microphysical properties with the observation will be helpful for readers to understand the accuracy of the model. It will be good to compare the vertical distribution of radar reflectivity from the model with observations. If available, surface precipitation can be compared to show the robustness of model simulations.

Reply: We appreciate your comment, and we totally agree that comparing the modeled microphysics to observation is helpful for readers to understand the accuracy of the model. Unfortunately, we do not have any direct measurements of cloud microphysics (such as airborne measurements), and the surface station data is not publicly available. The radar has no dual-frequency products to retrieve microphysics. To better understand the performance of the model, we plot the contoured-frequency-by-altitude diagram (CFAD) of reflectivity, which can statistically show the difference in the frequency distribution of reflectivity at different heights between observation and model

simulations. According to another reviewer's comments, a fourth secondary ice production mechanism by ice sublimational breakup has been added to our model (Deshmukh et al., 2022; Waman et al., 2022). The experiment with all four SIP processes included is named "4SIP". As seen in Fig. R1, the maximum reflectivity is observed at about 4 km, which is the height of the melting levels. The modeled maximum reflectivity from the noSIP experiment is larger than observed by about 7 dBZ, this is also seen from the map of composite reflectivity in the paper. With SIP implemented, the maximum reflectivity decreases and is more consistent with observation. The mean reflectivity profiles in both the noSIP and 4SIP experiments are systematically larger than observed, but the 4SIP performs better than noSIP experiment. The observed reflectivity may be underestimated at low levels (Fig. R2) because the lowest elevation angle used in the radar measurement is 0.5 degree (please see more information on measurements in reply to comment 7) and the low-elevation beams are affected by ground clutters.

[Figure]

Figure R1. The contoured-frequency-by-altitude diagram (CFAD) of reflectivity from (a-c) noSIP, (d-f) 4SIP experiments and (g-i) radar observation. The black lines indicate the profiles of mean reflectivity, and the dashed and dotted lines in (g-i) are the mean reflectivity profiles from noSIP and 4SIP experiments.

[Figure]

Figure R2. Observed radar reflectivity at (a) 500m and (b) 1000m a.m.s.l at 02:00, Nov. 28[th].

References:

Deshmukh, A., Phillips, V. T. J., Bansemer, A., Patade, S. and Waman, D.: New Empirical Formulation for the Sublimational Breakup of Graupel and Dendritic Snow, J. Atmos. Sci., 79(1), 317–336, doi:10.1175/JAS-D-20-0275.1, 2022.

Waman, D., Patade, S., Jadav, A., Deshmukh, A., Gupta, A. K., Phillips, V. T. J., Bansemer, A. and Demott, P. J.: Dependencies of Four Mechanisms of Secondary Ice Production on Cloud-Top Temperature in a Continental Convective Storm, J. Atmos. Sci., 79(12), 3375–3404, doi:10.1175/JAS-D-21-0278.1, 2022.

2. Even with radar reflectivity plots, contour levels are different in observations and model, which makes it difficult to compare. To what extent does the simulated radar reflectivity is in agreement with observations when all SIP processes are active? It will be good to present some statistical analysis.

Reply: Thank you for your comment. The contour levels are revised, and the composite reflectivity of the model and observation are combined in one figure (Fig. R3). It is seen that the composite radar reflectivity in the 4SIP experiment is more consistent with observation. The statistical comparison can be seen in the CFAD of the radar reflectivity (Fig. R1), the noSIP overestimates the maximum reflectivity by about 7 dBZ, and this uncertainty is reduced after implementing the SIP processes. Both the noSIP and 4SIP experiments systematically overestimate the reflectivity, because the occurrence frequency of reflectivity that is greater than 30 dBZ is higher in the model.

[Figure]

Figure R3. Composite radar reflectivity from (a-c) noSIP, (d-f) 4SIP experiment, and (g-i) observation at 02:00, 04:00, and 06:00, Nov 28th.

3. Based on my knowledge, in most of the previous studies, ice-ice collision is a major SIP mechanism in deep convective clouds when compared with rime-splintering and drop shattering (e.g. Phillips and Patade 2022). This is because rime-splintering and drop shattering are active over a limited range of temperatures. The authors need to mention the reasons behind less active ice ice collision in the simulated case. What are the factors that resulted in high secondary production by rime-splintering and drop shattering when compared with ice-ice collisions? What are the major differences in the microphysical processes of wintertime thunderstorms and summertime thunderstorms? There should be some discussion on the relative role of SIP in modulating ice number concentration and hence cloud electrification.

Reply: We appreciate your comment. The cloud electrification, in this case, is intensive around 00:00, when the graupel mixing ratio is high and the convection is deep (Fig. R4a). The enhancement of ice concentration by collisional breakup between ice crystals is mainly found after 01:00 when the ice/snow mixing ratio is high but the graupel mixing ratio is low (Fig. R5b and g). Cloud droplet concentration above the freezing level is also low after 01:00. Rime-splintering and drop shattering are active at relatively warm temperatures but throughout the entire life cycle. The relatively low graupel and droplet concentrations after 01:00 prevent the intensification of charge separation, this explains why collision between ice crystals has a weaker impact than rime-splintering and drop shattering on cloud electrification in this case. The ice-ice collisional breakup also enhances the ice concentration at temperatures warmer than -20 °C before 01:00, but the impact is weak due to the relatively low ice concentration in this period (Fig. R4g). Above -30 °C, the ice concentration is high but they are all small, thus the collisional breakup is insignificant. It is interesting to see that the ice concentration decreases in some areas (Fig. R5), in particular, the high-level ice concentration is reduced by more than 0.3 $L^{-1}$ due to the rime-splintering and drop shattering, this is because the SIP enhances the cloud glaciation at low levels, and the cloud droplets that transport to higher levels for freezing are reduced.

Though the impact of the ice-ice collisional breakup is relatively weak on average, it does not mean this SIP process only has a minor impact everywhere. Figure R6 shows the cross sections (black line in Fig. R3a) of the mixing ratio, in which we can see that ice-ice collisional breakup clearly enhances the ice/snow mixing ratio, and the charge separation is also enhanced (Fig. R7).

The SIPs may have different impacts in different cases. Huang et al. (2022) analyzed the relative contribution of 3 SIP processes to ice generation using model simulations, they compared the modeled microphysics to airborne observation, and the results show shattering of freezing droplets dominates ice particle production at temperatures between −15 and 0 °C during the developing stage of convection, and ice–ice collisional breakup dominates at temperatures during the later stage of convection. Studies that investigate the impacts of different SIPs on cloud electrification are still limited. It will be interesting to see how changes in different environmental conditions (such as wind shear, cloud base height, and aerosol and INP concentrations) in different cases would

influence the role of different SIPs. Based on this study, it is suggested that sufficient graupel is important for SIP processes to enhance cloud electrification. This discussion is added to the revised paper.

In many previous studies of summertime thunderstorms that occurred at a similar latitude (e.g., Caicedo et al., 2018; Shi et al., 2015), the main charging region is typically at 5-11 km a.m.s.l., and the freezing level is at about 5 km a.m.s.l, which are all about 1 km higher than the cold-season storm shown in this paper. Kitagawa and Michimoto (1994) found that wintertime thunderstorms generally have shorter periods of electrical activity and less frequent lightning than summer thunderstorms. In addition, they noted that the tropopause remains at 16 km in summer and drops to 10 km in winter, and the vertical extent of the atmospheric circulation is about half that of summer. This is the main factor that limits the convective activity of thunderstorm clouds in winter. Takahashi et. al. (2017) pointed out that winter thunderstorm clouds have lower liquid water content (LWC) and low cloud tops. In our simulation, the modeled LWC is typically lower than 1 g m$^{-3}$, which is lower than that reported in summer convections (e.g., Yang et al., 2016; Phillips et al. 2022). The lower LWC in wintertime convection indicates weaker riming, thus a lower riming accretion rate, which potentially leads to a higher occurrence of inverted charge structure of thunderstorms (Wang et al. 2021). This discussion is added to the revised paper.

[Figure]

Figure R4. Temporal evolution of (a, e) graupel/hail, (b, f) ice/snow, (c, g) rain, and (d,

h) cloud drop mixing ratio and concentration in the noSIP experiment.

[Figure]

Figure R5. Differences in the number concentration of different hydrometeors between the experiments with SIP and those without SIP. (a, f, k, p) experiment with rime-splintering, (b, g, l, q), experiment with ice-ice collisional breakup (c, h, m, r) experiment with shattering of freezing drops, (d, i, n, s) experiment with ice breakup during sublimation, and (e, j, o, t) experiment with four SIP processes.

[Figure]

Figure R6: Cross sections of the modeled mixing ratio for (a)-(f) graupel/hail, (g)-(l)

snow/ice, (m)-(r) rain and (s)-(x) cloud droplet at 01:00, Nov. 28[th].

[Figure]

Figure R7: Cross sections of the modeled (a-f) graupel charge density and (g-l) noninductive charging rate at 01:00, Nov. 28[th].

References:

Caicedo, J. A., Uman, M. A., and Pilkey, J. T. : Lightning Evolution In Two North Central Florida Summer Multicell Storms and Three Winter/Spring Frontal Storms. Journal of Geophysical Research: Atmospheres, 123(2), 1155–1178, https://doi.org/10.1002/2017JD026536, 2018.

Huang, Y., Wu, W., McFarquhar, G. M., Xue, M., Morrison, H., Milbrandt, J., Korolev, A. V., Hu, Y., Qu, Z., Wolde, M., Nguyen, C., Schwarzenboeck, A., and Heckman, I.: Microphysical processes producing high ice water contents (HIWCs) in tropical convective clouds during the HAIC-HIWC field campaign: Dominant role of secondary ice production. Atmospheric Chemistry and Physics, 22(4), 2365–2384, https://doi.org/10.5194/acp-22-2365-2022, 2022.

Kitagawa, N., and Michimoto, K.: Meteorological and electrical aspects of winter thunderclouds. Journal of Geophysical Research, 99(D5), 10713, https://doi.org/10.1029/94JD00288, 1994.

Phillips, V. T. J., and Patade, S.: Multiple Environmental Influences on the Lightning of Cold-Based Continental Convection. Part II: Sensitivity Tests for Its Charge Structure and Land–Ocean Contrast. Journal of the Atmospheric Sciences, 79(1), 263–300, https://doi.org/10.1175/JAS-D-20-0234.1, 2022.

Shi, Z., Tan, Y. B., Tang, H. Q., Sun, J., Yang, Y., Peng, L., and Guo, X. F.: Aerosol

effect on the land-ocean contrast in thunderstorm electrification and lightning frequency. Atmospheric Research, 164–165, 131–141, https://doi.org/10.1016/j.atmosres.2015.05.006, 2015.

Takahashi, T., Sugimoto, S., Kawano, T., and Suzuki, K.: Riming Electrification in Hokuriku Winter Clouds and Comparison with Laboratory Observations. Journal of the Atmospheric Sciences, 74(2), 431–447, https://doi.org/10.1175/JAS-D-16-0154.1, 2017.

Yang, J., Wang, Z., Heymsfield, A., and Luo, T.: Liquid-ice mass partition in tropical maritime convective clouds. J. Atmos. Sci., 73, 4959-4978, doi: 10.1175/JAS-D-15-0145.1, 2016.

Wang, D., Zheng, D., Wu, T., and Takagi, N.: Winter Positive Cloud‐to‐Ground Lightning Flashes Observed by LMA in Japan. IEEJ Transactions on Electrical and Electronic Engineering, 16(3), 402–411, https://doi.org/10.1002/tee.23310, 2021.

4. It is important to show the rates of three SIP processes implemented in the model. Or at least the concentration of ice resulting from each SIP mechanism in 3SIP simulations can be shown. Time height evolution of ice particle number concentration from each of the SIP mechanisms will help to understand their relative importance in altering total ice number concentration. Authors have shown time height evolution of mass mixing ratios, however, changes in ice number concentration are very important as far as the role of SIP is concerned.

Reply: Thank you for your comment. The time height evolution of changes in ice number concentration is shown in Fig. R5. The ice production rate of the four SIP processes in the 4SIP experiment is better illustrated using cross sections, as the locations where secondary ice production is intense are different among the four processes. As seen in Fig. R8, the rime-splintering and drop shattering produce significant secondary ice in the core of clouds, where the graupel and rain mixing ratio are high, while the sublimational breakup of ice/snow is more intense near cloud edges or regions with relatively low reflectivity, probably because of the entrainment mixing and regional downdrafts. Ice-ice collisional breakup is more intense in regions with high ice/snow concentrations (Fig. R8b and R6l). The ice production rate by rime-splintering is the highest, and that by the sublimational ice breakup is the lowest, this

substantial difference in the magnitude of the ice production rate is also true after averaging the entire cloud region, and it explains why the rime-splintering process has the most significant impact on the cloud microphysics.

[Figure]

Figure R8: Cross section of the production rates of the secondary ice resulting from four SIP mechanisms. (a) experiment with rime-splintering (RS), (b) experiment with ice-ice collisional breakup (IC), (c) experiment with shattering of freezing drops (SD), (d) experiment with sublimational breakup (SK).

5. In Figure 8, temporal variation of ice/snow showed that there is not much effect of individual SIP process on ice/snow concentration, however when all SIPs were considered the concentration was boosted. What are the physical mechanisms behind it? I expect a significant increase in ice/snow concentration as a result of SIP in the simulations where a single SIP is considered if that mechanism is important.

Reply: Thank you for your comment. The original Fig. 8 was a display of averages over area and time, which lost some detailed information. Now we average only in cloud region. We can see that a single SIP process, especially by RS and SD, has clear influences on cloud microphysics (Fig. R5).

6. A few details of the implementation of SIP in the model are needed. What was the diameter of the tiny fragments that resulted from mode 1 in drop shattering? What kind of collisions were considered for collisional breakup mechanisms? In which category

the resulting fragments were added?

Reply: Thank you for your comment. Detailed information has been added in Appendix A of the manuscript. In our model, the tiny fragments are treated as the ice particles belonging to the first bin of the Fast-SBM scheme, which has a diameter of 4 micrometers (Khain et al., 2004). The collision between ice/snow and ice/snow has been considered in this paper. In this paper, except for the tiny fragments mentioned above, other fragments are added to the mass bins of the Fast-SBM scheme coinciding with their mass.

Reference:

Khain, A., Pokrovsky, A., Pinsky, M., Seifert, A., and Phillips, V.: Simulation of Effects of Atmospheric Aerosols on Deep Turbulent Convective Clouds Using a Spectral Microphysics Mixed-Phase Cumulus Cloud Model. Part I: Model Description and Possible Applications. Journal of the Atmospheric Sciences, 61, 2963–2982, doi: 10.1175/JAS-3350.1, 2004.

7. There is no information about the radar data e.g. which radar was used, what are the data corrections etc. Similarly, there is not much information available about lightning data.

Reply: We appreciate your comment. The following information is added to the paper. *"Radar reflectivity can be used to illustrate the intensity of the storm. The radar data used in this study is a gridded product generated based on 32 S-band radars operated across southeast China. For each radar, the detection radius is 230 km, the range resolution is 250 m and the beamwidth is 1°. The radar finishes a volume scan every 6 minutes consisting of 9 elevation angles (0.5°, 1.5°, 2.4°, 3.4°, 4.3°, 6.0°, 9.9°, 14.6° and 19.5°). The data recorded by these radars were interpolated into a Cartesian grid with a horizontal resolution of 1 km and vertical resolution of 500 m based on the Cressman technique.*

*In addition, the lightning location and flash rate are evaluated using observation. The lightning location data is obtained based on the very low frequency (VLF) lightning location network (LLN) in China developed by Nanjing University of Information Science and Technology (Li et al., 2022). VLF-LLN was established in 2021 and has 26 stations distributed across various regions in China. The detection area covers the*

*entire China as well as parts of East and Southeast Asia. The lightning location algorithm is developed based on the time-of-arrival (TOA) method, and the arrival times of each lighting-induced pulse at different stations are obtained by matching the recorded waveforms to the idealized waveforms simulated using the Finite Difference Time-Domain (FDTD) technique. The lightning location error is 1-5 km (Li et al., 2022).*

*Moreover, the NCEP reanalysis data is used to investigate the synoptic conditions, the sounding measurements at Fuyang, which is conducted every 12 hours, is used to investigate the thermodynamic conditions, and the brightness temperature (TBB) on FY2H satellite that is developed in China is used to illustrate the cloud coverage.*

Reference:

Li, J.; Dai, B.; Zhou, J.; Zhang, J.; Zhang, Q.; Yang, J.; Wang, Y.; Gu, J.; Hou, W.; Zou, B, and Li, J.: Preliminary Application of Long-Range Lightning Location Network with Equivalent Propagation Velocity in China. Remote Sens. 14, 560, doi: 10.3390/rs14030560, 2022.

8. Line 378: if ice ice collision was less active what are the reasons behind the enhancement in the flash rate?

Reply: We appreciate your comment. In the original paper, the enhancement of flash rate by ice-ice collision is found between 22:00 Nov. 27th and 00:00 Nov. 28th. During this period, the graupel mixing ratio is enhanced (Fig. 7 in the original paper), and cloud electrification is enhanced by ice-ice collision (Fig. 12 in the original paper). In the revised paper, the parameters used in the ice-ice collisional breakup scheme are updated based on Gautam (2022). The result is different from the original one. As seen in Fig. R9, the flash rate has a similar magnitude in noSIP and IC experiments. In addition, as discussed in comment 3, ice-ice collisional breakup could be more important than the other SIP processes in some areas (Fig. R6 and R7). It is interesting to see such a significant difference in the original and updated IC experiments, suggesting it is important to use correct parameters in the scheme.

[Figure]

Figure R9. The (a) lightning location, and (b) time series of flash rate from observation and the six numerical experiments.

Reference:

Gautam, M. Fragmentation in graupel snow collisions. Master of Science dissertation, Dept of Physical Geography and Ecosystem Science, Lund University, Lund, Sweden, DOI: http://lup.lub.lu.se/student-papers/record/9087233, 2022.

9. What are the mechanisms behind the improvement in the temporal distribution of flash rate in 3SIP simulations?

Reply: Thank you for your comment. According to Fig. R5, it is seen that the graupel mixing ratio (and concentration) is enhanced throughout the cloud life cycle, and the

electric field is enhanced after 00:00, Nov 28[th]. Rime-splintering is the main mechanism that leads to the enhancement. This can be interpreted based on the equation of non-inductive charging produced during the collision between graupel and ice crystal:

$$\frac{\partial \rho_{gi}}{\partial t} = \iint_0^\infty \frac{\pi}{4} \beta \delta q_{gi} (1 - E_{gi}) |V_g - V_i| (D_g + D_i)^2 n_g n_i dD_g dD_i$$

Based on this equation, we can see the charge transfer is determined by three terms: 1) charge transferred during each collision between graupel and ice ($\delta q_{gi}$); 2) collision kernel between graupel and ice; 3) concentration of graupel and ice. $\delta q_{gi}$ is determined by RAR, which is a function of liquid water content (LWC) and terminal velocity of graupel. With the addition of SIP, the LWC generally decreases (Fig. R5), and the diameters of ice particles decrease as well, leading to a decrease in RAR (Fig. R10), especially in RS and SD experiments. The collision kernel between graupel and ice is determined by the terminal velocity and size of graupel and ice, which also decreases after SIP processes are implemented. The concentration of graupel ($n_g$) and ice ($n_i$) increases due to the RS and SD processes, this explains the enhanced electrification by these two processes. Therefore, the higher graupel and ice concentration induced by RS and SD processes is the main reason resulting in the enhanced electric field, which leads to more flash rate after 00:00, Nov 28[th].

[Figure]

Figure R10: The time height averaged diagram of RAR. (a) experiment without SIP, (b) experiment with rime-splintering (RS), (c) experiment with ice-ice collisional breakup (IC), (d) experiment with shattering of freezing drops (SD), (e) experiment with sublimational breakup (SK) and (f) experiment with four SIP processes.

10. Authors should check the manuscript carefully for grammar and language corrections. In many places, articles are missing or not used properly.

Reply: We appreciate your comment and are sorry for the language errors. In the revised paper, we have carefully read through the manuscript and corrected grammatical errors.

Technical corrections/Minor comments:

1. What was the cloud base height and temperature of simulated clouds?

Reply: We do not have direct measurements of cloud base height. According to the sounding measurement and model simulation, the cloud base height and temperature are about 1 km and 12 ℃.

2. Figure 1 captions: The time mentioned in the caption does not match that mentioned in the plots. Also, plots 1a and 1b are supposed to be 500 mb geopotential height, isotherms and wind barbs, but on plot b the mentioned height is 850 hpa. The same mistake is with plots c and d.

Reply: Thank you for your comment. This mistake has been corrected.

3. Figure 7: What are the averaging conditions for incloud points shown in time height plots?

Reply: We apologize for not explaining it clearly. The original figure in the manuscript shows the domain average time-height variation. In the revised paper, we only average the in-cloud area, which is identified using a total water mixing ratio (including all hydrometeors) greater than $10^{-6}$ g/kg.

4. Figure 7 captions: The names of sensitivity studies mentioned in the captions "SBM-0SIP Simulation; SBM-1SIP SimulationSBM-2SIP Simulation; SBM-3SIP" do not match the names on the plot. Please correct it according to the sensitivity tests mentioned in the text earlier.

Reply: Thank you for your comment and sorry for the mistake. It has been corrected in the revised paper.

Line 11: in a thunderstorm that occurred;

Reply: Thank you for your comment. This sentence is revised to "… in a thunderstorm that occurred …"

Line 11: are investigated …

Reply: Thank you for your comment. "is" is changed to "are".

Line 40: correct lighting to lightning

Reply: Thank you for your comment. "lighting" is changed to "lightning".

Line 55: Phillips et al. 2020

Reply: Thank you for your comment and sorry for the typo. It is corrected in the paper.

Line 67: studies that highlighted ….

Reply: Thank you for your comment, "that" is added in the sentence.

Line 95: warm moist ..

Reply: Thank you for your comment. "most" is changed to "moist".

Line 111: Fig.3a not 2a

Reply: Thank you for your comment, and the figure number is corrected accordingly.

Line 113: Fig 3c not 2c

Reply: Thank you for your comment, and the figure number is corrected accordingly.

Line 124: Figure 4 not 3

Reply: Thank you for your comment, and the figure number is corrected accordingly.

Line 124: a two-way nested

Reply: Thank you for your comment, "a" is added before "two-way".

Line 127: spin-up

Reply: Thank you for your comment. "spin up" is changed to "spin-up".

Line 140: Incomplete sentence

Reply: Thank you for your comment. This sentence is changed to "At temperatures colder than -8 °C or warmer than -3 °C, the rime-splintering is inactive."

Line 140: at temperatures colder than

Reply: Thank you for your comment. "temperature" is changed to "temperatures".

Line 144: it can also be active …

Reply: Thank you for your comment. "be" is added in the sentence.

Line 204: change "With all the three SIP processes implement" to "With all implemented"

Reply: Thank you for your comment. "With all the three SIP processes implement" is changed to "With all implemented".

Line 217: units should be g kg-1 and not g ks-1

Reply: Thank you for your comment. g ks$^{-1}$ is changed to g kg$^{-1}$.

Line 232: there are …

Reply: Thank you for your comment. "there are" is changed to "there is".

Line 235: graupel mixing ratio …

Reply: Thank you for your comment. "ratio" is added in the sentence.

Line 245: correct quicky to quickly

Reply: Thank you for your comment. "quicky" is changed to "quickly".

Line 281: results in changes in the

Reply: Thank you for your comment. "result" is changed to "results".

Line 318: delete the before that

Reply: Thank you for your comment. "the" is deleted.

Line 329: cross-section

Reply: Thank you for your comment. "cross section" is changed to "cross-section".

Line 380: implemented

Reply: Thank you for your comment. "implemented" is revised.

Line 407: replace continued by continue

Reply: Thank you for your comment. "continued" is changed to "continue".

Line 408: change falling to it falls

Reply: Thank you for your comment. "falling" is changed to "it falls".

Line 414: change on to in

Reply: Thank you for your comment. "on" is changed to "in".

Line 469: Define RAR and RARc

Reply: Thank you for your comment. RAR (riming accretion rate) and RARc (critical riming accretion rate) are defined in the revised paper.

**Reviewer 2:、**

The impacts of three secondary ice production (SIP) processes on the electrification were evaluated by the mesoscale simulation. A new electrical model was constructed based on the fast spectral bin microphysics (SBM) scheme, which constitutes a significant contribution of this paper. This electrical model will serve as an effective tool for studying electrification and discharge processes. However, there are several key issues in the paper that require further clarification. Substantial revisions may be necessary to strengthen the supporting evidence. Specifically, the following matters should be considered:

Reply: We appreciate your insightful comments. The paper has been revised accordingly and has been improved a lot. Please see our responses below.

1. In the model validation section, it is recommended that the author can display a two-dimensional distribution of observed and simulated lightning activities. The entire inner domain is too large to effectively reflect the distribution of simulated lightning activity.

Response: Thank you for your comment. The two-dimensional distribution of observed and simulated lightning activities is shown in Fig. R11 and added in the revised paper. The location of lightning is consistent with low brightness temperature. In the model, we failed to simulate the convective cell between 30°N and 32°N, the modeled reflectivity in this area is lower than the observed. While for the southern convective cell, the simulations are consistent with observation. According to another reviewer's comments, a fourth secondary ice production mechanism by ice sublimational breakup has been added to our model (Deshmukh et al., 2022; Waman et al., 2022). The experiment with all four SIP processes included is named "4SIP", and that with ice sublimational breakup only is named "SK".

Reference:

Deshmukh, A., Phillips, V. T. J., Bansemer, A., Patade, S. and Waman, D.: New Empirical Formulation for the Sublimational Breakup of Graupel and Dendritic Snow, J. Atmos. Sci., 79(1), 317–336, doi:10.1175/JAS-D-20-0275.1, 2022.

Waman, D., Patade, S., Jadav, A., Deshmukh, A., Gupta, A. K., Phillips, V. T. J., Bansemer, A. and Demott, P. J.: Dependencies of Four Mechanisms of Secondary Ice Production on Cloud-Top Temperature in a Continental Convective Storm, J.

Atmos. Sci., 79(12), 3375–3404, doi:10.1175/JAS-D-21-0278.1, 2022.

[Figure]

Figure R11. The (a) lightning location, and (b) time series of flash rate from observation and the six numerical experiments.

2. How was Figure 7 (as well as all time-height diagrams) created? Does Figure 7 present single-point data or regional average data? If it is regional average data, is it an average of the entire inner domain? Time-height diagrams for single-point or regional average data may better display the changing trends of variables, while cross sections can provide a more intuitive understanding. Cross sections for charging rates could also be shown.

Response: We appreciate the comment, and sorry for not explaining clearly. In the

original paper, Figure 7 shows an average of the entire inner domain. In the revised paper, we average only in-cloud regions. The cross sections of mixing ratio and number concentration of graupel, snow, and rain are shown in Fig. R12 and R13, and the cross sections of charge density on graupel and noninductive charging rate are shown in Fig. R14. These figures help understand the spatial distribution of microphysics and electrification. It is seen that though the impact of the ice-ice collisional breakup is small on average (which is seen in the time-height diagrams), this process can significantly enhance graupel or ice concentration in some areas (Fig. R12c, i, o). It is difficult to tell which SIP process has the most significant impact on cloud microphysics simply based on these cross sections, as the composite impact of the four SIP processes is not simply a sum of them (Fig. R12f, l, r, x). But according to the time-height diagrams, on average, the rime-splintering has a stronger impact on the cloud microphysics and electrification (Fig. R16)

The charge separation is found in areas with relatively high graupel and ice concentration. All four SIP processes, especially the rime-splintering process, can enhance positive charge separation at low levels. In addition, the ice-ice collisional breakup can enhance negative charging rates at high levels. The graupel charge density in the 4SIP experiment is more similar to that in the RS experiment. These cross-sections also help to understand the substantial difference between the charge density and charging rate, which is related to comment 3 (please see reply to comment 3.)

[Figure]

Figure R12: Cross sections of the modeled mixing ratio for (a)-(f) graupel/hail, (g)-(l)

snow/ice, (m)-(r) rain and (s)-(x) cloud droplet at 01:00, Nov. 28th.

[Figure]

Figure R13: The same as Fig. R12 but for concentration.

[Figure]

Figure R14: Cross sections of the modeled (a-f) graupel charge density and (g-l) noninductive charging rate at 01:00, Nov. 28th.

3. The correlation between charge structure and electrification rate does not align. Based on the charging rate distribution presented in Figure 9, it is observed that although the inductive charging rate varies significantly across different experiments, the non-inductive charging rate is one order of magnitude higher than the inductive charging rate. Therefore, we continue to lean towards the notion that electrification in the cloud is primarily attributed to the non-inductive collision process. However, it is

noteworthy that even when there is a minimal difference in the non-inductive electrification rate (Fig. 9 a, c, e, g, i), it leads to completely distinct charge distributions (Fig. 10), which is indeed perplexing. If our understanding is correct, the non-inductive charging rate depicted in Figure 9 should be targeted at graupel particles. Given this distribution, it should not cause such a considerable difference in electrification as observed in Figure 10. However, the substantial difference is difficult to explain solely by the sedimentation of graupel particles. Is it possible that the regional average has concealed some crucial information? Or we suspect that the inductive charging process may also play a vital role in the formation of charge structure. Therefore, it is suggested that the effects of inductive and non-inductive electrification should be separated.

Response: Thank you for your professional comment. According to your comment, we made a sensitivity test in which only noninductive electrification is used. Figure R15 shows the graupel charge density, noninductive charging rate, and the fraction of area with charge separation occurring. In the noSIP experiment, the graupel charge density is negative, while the noninductive charging rate has a bipolar structure. The magnitude of the low-level positive charging rate is much smaller than the high-level negative charging rate. This result is the same as that shown in the original paper, in which both noninductive and inductive charging are considered. Therefore, it is evident that the charge density is mainly controlled by noninductive charging. The different structures of the average charge density and charging rate indicate some crucial information is canceled by averaging. Since a threshold of RAR>0.1 g m$^{-3}$ s$^{-1}$ is required to trigger charge separation, charging takes place only in a small fraction of the cloud area (Fig. R15e and f). This is more intuitive in the cross sections shown in Fig. R14. Charging only occurs in areas with relatively high graupel concentration (Figs. R14g-l and R3), while fall of graupel with negative charge is found in more areas. If the magnitude of the low-level positive charging rate is small, the average charge density would be negative, while if the magnitude of the low-level positive charging rate is enhanced by SIP, the average low-level charge density on graupel is positive. This information is added to the paper.

[Figure]

Figure R15. Time height diagrams of (a, b) graupel charge density, (c, d) noninductive charging rate, and (e, f) fraction of area with charge separation occurring in noSIP and RS experiments with only noninductive charging used.

4. The rationale for the charge structure differences in various experiments is not clear. Although the author demonstrated the differences in charge structure caused by different SIP processes, we believe that the underlying reason has not been fully disclosed. When the SIP process changes, what is the fundamental alteration? Which leads to the change in electrification rate and charge structure? Has the rime accretion rate (RAR) changed significantly? What causes the change in RAR?

Response: Thank you for your professional comment. The most significant change in charge structure is the low-level positive charge is enhanced, especially by RS. This can be interpreted based on the equation of non-inductive charging produced during the collision between graupel and ice crystal:

$$\frac{\partial \rho_{gi}}{\partial t} = \iint_0^\infty \frac{\pi}{4} \beta \delta q_{gi} (1 - E_{gi}) |V_g - V_i| (D_g + D_i)^2 n_g n_i dD_g dD_i$$

Based on this equation, we can see the charge transfer is determined by three terms: 1) charge transferred during each collision between graupel and ice ($\delta q_{gi}$); 2) collision kernel between graupel and ice; 3) concentration of graupel and ice. $\delta q_{gi}$ is determined by RAR, which is a function of liquid water content (LWC) and terminal velocity of graupel. With the addition of SIP, the LWC generally decreases (Fig. R16),

and the diameters of ice particles decrease too, leading to a decrease in RAR (Fig. R17), especially in RS and SD experiments. The collision kernel between graupel and ice is determined by the terminal velocity and sizes of graupel and ice, which also decreases after SIP processes are implemented. The concentrations of graupel ($n_g$) and ice ($n_i$) increase due to the RS and SD processes, this explains the enhanced low-level charging by these two processes. This is added in the discussion section.

[Figure]

Figure R16. The difference in mixing ratio of (a-e) graupel/hail, (f-j) ice/snow, (k-o) rain and (p-t) cloud droplets between the experiments with SIP and that without SIP.

[Figure]

Figure R17: The time height averaged diagram of RAR. (a) experiment without SIP, (b)

experiment with rime-splintering (RS), (c) experiment with ice-ice collisional breakup (IC), (d) experiment with shattering of freezing drops (SD), (e) experiment with sublimational breakup (SK) and (f) experiment with four SIP processes.

5. The author should illustrate the specific location of the cross-sections in Figure 12 within Figure 5.

Response: Sorry for not explain clearly. The location of the cross-sections is illustrated in the revised figure now (black line in Fig. R18a).

[Figure]

Figure R18. Composite radar reflectivity from (a-c) noSIP, (d-f) 4SIP experiment, and (g-i) observation at 02:00, 04:00, and 06:00, Nov 28th.

**Reviewer 3:**

**General comments:**

Yang et al. investigate the role of three secondary ice production (SIP) processes in precipitation intensity, cloud electrification, and discharge processes, within the context of a wintertime thunderstorm. The analysis relies on mesoscale simulations conducted using the Weather Research and Forecasting (WRF) model, coupled with a fast spectral bin microphysics (SBM) scheme. The employed SBM scheme was refined through the incorporation of state-of-the-art ice multiplication formulations complemented by the integration of noninductive and inductive charging parameterizations.

This study contributes significantly to clarifying the complex interactions between ice microphysics – particularly the poorly constrained SIP processes – and cloud electrification. Despite its importance, the manuscript requires substantial revisions across the methodology, model evaluation and results sections, aimed at improving readability and enhancing the robustness of certain findings. It is recommended that the following aspects be revisited before publication:

Reply: We appreciate your insightful comments. The paper has been revised accordingly and has been improved a lot. Please see our responses below.

**Specific comments:**

1. In Section 2.2, I would recommend to explain the rationale behind selecting the specific microphysics scheme, specifying the ice and liquid hydrometeor species considered in the model, and providing information on whether this scheme has undergone evaluation in similar studies in the past.

Reply: Thank you for your comment. The following information is added to the revised paper.

*"Compared to bulk microphysics scheme, spectral bin microphysics (SBM) scheme has the advantage of calculating particle size distributions (PSDs) by solving explicit microphysical equations. It aims to simulate as accurately as possible cloud microphysical processes (Khain et al. 2015). In the fast version of SBM in WRF, the ice and liquid hydrometeor species include cloud droplet/rain, ice/snow, and graupel, each of them is represented by 33 doubling mass bins. It has been demonstrated that SBM performs better than bulk microphysics in modeling cloud microphysics in many previous studies (e.g., Fan et al., 2012; Khain et al. 2015). However, SBM has not been*

*widely used for studying cloud electrification (e.g., Mansell et al., 2005; Shi et al., 2015). Recently, Philips et al. (2020) implemented cloud electrification parameterization in the SBM in a cloud model, and they conducted an idealized simulation of deep convective clouds. The results showed the modeled charge structure and lightning activity are consistent with observations. However, cloud electrification has not been implemented in SBM in WRF for real case study before."*

References:

Fan, J., L. R. Leung, Z. Li, H. Morrison, H. Chen, Y. Zhou, Y. Qian, and Y. Wang: Aerosol impacts on clouds and precipitation in eastern China: Results from bin and bulk microphysics. J. Geophys. Res., 117, D00K36, doi:10.1029/2011JD016537, 2012.

Mansell, E. R., MacGorman, D. R., Ziegler, C. L., and Straka, J. M.: Charge structure and lightning sensitivity in a simulated multicell thunderstorm. Journal of Geophysical Research, 110, D12101, doi: 10.1029/2004JD005287, 2005.

Khain, A. P., et al. : Representation of microphysical processes in cloud- resolving models: Spectral (bin) microphysics versus bulk parameterization. Rev. Geophys., 53, 247–322, doi:10.1002/2014RG000468, 2015.

Phillips, V. T., Formenton, M., Kanawade, V. P., Karlsson, L. R., Patade, S., Sun, J., Barthe, C., Pinty, J. P., Detwiler, A. G., Lyu, W. and Tessendorf, S. A.: Multiple environmental influences on the lightning of cold-based continental cumulonimbus clouds. Part I: Description and validation of model. J. Atmos. Sci., 77, 3999-4024, doi: 10.1175/JAS-D-19-0200.1, 2020.

Shi, Z., Tan, Y. B., Tang, H. Q., Sun, J., Yang, Y., Peng, L., and Guo, X. F.: Aerosol effect on the land-ocean contrast in thunderstorm electrification and lightning frequency. Atmospheric Research, 164–165, 131–141, doi: 590 10.1016/j.atmosres.2015.05.006., 2015.

2. Regarding the implementation of the ice-ice collisional break-up (IC) and the shattering of freezing drops (SD), it is important to provide a more detailed description – especially if this is the first attempt to incorporate these parameterizations into the SBM scheme:

Reply: Thank you for your valuable reminding. The detailed information is added in Appendix A of the revised paper as follows.

*"The parametrization of ice–ice collisional breakup is developed by Phillips et al.*

*(2017). The number of ice fragments produced during ice–ice collision is:*

$$N_{IC} = \alpha A(M) \left\{ 1 - exp \left[ - \left( \frac{C(M)K_0}{\alpha A(M)} \right)^{\gamma} \right] \right\} \tag{A3}$$

*where $A(M)$ is the number density of breakable asperities on the ice particle and related to the rimed fraction and the size of smaller ice particle, $C(M)$ is asperity– fragility coefficient that is set as $3.86 \times 10^4$ according to the cloud chamber experiment of natural ice particles (Gautam, 2022), $K_0$ is the initial value of collision kinetic energy, $\gamma$ and $\alpha$ are the shape parameter and the equivalent spherical surface area of smaller particles, respectively. $\gamma = 0.5 - 0.25\Psi$, where $\Psi$ denotes the rimed fraction, which is assumed 0.2 in this study. The tiny fragments are treated as the ice particles belonging to the first bin of the Fast-SBM model.*

*The parameterization of shattering of freezing drops was developed by Phillips et al. (2018) based on laboratory experiments. If contact with a smaller ice particle, a supercooled drop may break and produce both big and tiny ice fragments, thus, the number of the ice fragments can be expressed using:*

$$N_{SD\_1} = N_T + N_B \tag{A4}$$

$$N_{SD\_1} = F(D)\Omega(T) \left[ \frac{\xi_T \eta_T^2}{\left( T - T_{T,0} \right)^2 + \eta_t^2} + \beta T \right] \tag{A5}$$

$$N_B = min \left\{ F(D)\Omega(T) \left[ \frac{\xi_B \eta_B^2}{\left( T - T_{B,0} \right)^2 + \eta_B^2} \right], N_T \right\} \tag{A6}$$

*where, $N_T$ and $N_B$ are the number of tiny and big ice fragments generated by a shattered drop. $F(D)$ and $\Omega(T)$ are the interpolating functions for the onset of drop shattering. $\xi_T$, $\xi_B$, $\eta_T$, $\eta_B$, $T_{T,0}$, $T_{B,0}$, $\beta$, are parameters determined based on datasets from previous laboratory experiments, which can be found in Phillips et al. (2018). The tiny fragments are treated as the ice particle belonging to the first bin of Fast-SBM model, which have a diameter of 4 micrometers (Khain et al., 2004). The mass of big ice fragments is $m_B = 0.4m_{drop}$.*

*In addition, a drop may also break if contacting with a more massive ice particle. The number of ice fragments produced in this process is:*

$$N_{SD\_2} = 3\Phi \times [1 - f(T)] \times max \left\{ \left( \frac{k_0}{S_e} - DE_{crit} \right), 0 \right\} \tag{A7}$$

$$f(T) = \frac{-C_w T}{L_f} \tag{A8}$$

$$S_e = \gamma_{liq}\pi D^2 \qquad\qquad (A9)$$

*where, $\gamma_{liq}$ is the surface tension of liquid drop, $k_0$ is the initial kinetic energy of the two colliding particles, $f(T)$ is the frozen fraction. $C_w$ and $L_f$ are the specific heat capacity of water and the specific latent heat of freezing, respectively. $DE_{crit} = 0.2$, and $\Phi$ is 0.3 according to James et al. (2021). All ice fragments are assumed to be tiny in this mode. The tiny ice fragments are added to the first bin of ice size distribution."*

References:

Gautam, M.: Fragmentation in graupel snow collisions. Master of Science dissertation, Dept of Physical Geography and Ecosystem Science, Lund University, Lund, Sweden-, doi: http://lup.lub.lu.se/student-papers/record/9087233, 2022.

James, R. L., Phillips, V. T. and Connolly, P. J.: Secondary ice production during the break-up of freezing water drops on impact with ice particles. Atmospheric Chemistry and Physics, 21, 18519-18530, doi: 10.5194/acp-21-18519-2021, 2021.

Khain, A., Pokrovsky, A., Pinsky, M., Seifert, A., and Phillips, V.: Simulation of Effects of Atmospheric Aerosols on Deep Turbulent Convective Clouds Using a Spectral Microphysics Mixed-Phase Cumulus Cloud Model. Part I: Model Description and Possible Applications. Journal of the Atmospheric Sciences, 61, 2963–2982, doi: 10.1175/JAS-3350.1, 2004.

Phillips, V. T. J., Yano, Jun-Ichi, Khain, A.: Ice Multiplication by Breakup in Ice-Ice Collisions. Part I: Theoretical Formulation. J. Atmos. Sci., doi: 10.1175/JAS-D-16-0224.1, 2017.

Phillips, V. T. J., Patade, S., Gutierrez, J., and Bansemer, A.: Secondary Ice Production by Fragmentation of Freezing Drops: Formulation and Theory. J. Atmos. Sci., 75, 3031–3070, doi: 10.1175/JAS-D-17-0190.1, 2018.

- The physically-based parameterization of Phillips et al. (2017) explicitly considers the effect of ice habit, ice type and rimed fractions of the particles undergoing fragmentation. These parameters are not always described in models, and therefore certain assumptions have to be made. Please describe how these parameters are treated in the model and whether the scheme predicts the rimed mass fraction of colliding ice particles or if a constant value is prescribed. Given the demonstrated impact of the rimed fraction on the efficiency of the IC mechanism (e.g., Karalis et al., 2022;

Sotiropoulou et al., 2021), you may consider assessing the sensitivity of your results to this parameter. Additionally, further clarification is needed regarding the collection efficiencies of ice particles and whether all collisions between ice particles can lead to fragmentation and the generation of SIP particles.

Reply: Thank you for your comment. The parameterization of Phillips et al. (2017) has more detailed physical processes than ours. In WRF fast-SBM, cloud ice is divided into high-density graupel and low-density ice/snow. This model does not distinguish between ice and snow, nor does it differentiate between graupel and hail (Khain et al., 2009). This is now clarified in the revised paper.

In our model, the rimed fraction is set as 0.2. The results of sensitive experiments for different rimed fraction values (0.2 and 0.4) are shown in Fig. R19 to Fig. R22. Figures R19 and R20 show the mixing ratio and number concentration for different rimed fractions. It can be seen that with a rimed fraction of 0.4, there are more graupel particles before 02:00, Nov. 28[th]. The mixing ratio and concentration of snow particles is also enhanced. As shown in Fig. R21, a larger rimed fraction leads to a slight increase in the charge density on graupel and snow. For the total charge density (Fig. R22), the upper-level negative charge and middle-level positive charge region is enhanced. This is added in the discussion section in the revised paper.

The collection efficiency of ice/snow is the product of collision efficiency and coalescence efficiency. The collision efficiency is obtained based on the Bohm's theory (Bohm, 1992a, 1992b) and the superposition method in Khain et al. (2001). The coalescence efficiency is parameterized in Khain and Sednev (1996), which can be expressed using:

$$E_{coal} = \min \left[1, \frac{e}{e_i} \max \{0, a + bT_c + cT_c^2 + dT_c^3\}\right]$$

where $e$ is the vapor pressure, $e_i$ is the saturation vapor pressure with respect to ice, $a, b, c$ are constant coefficients. As described in Phillips et al. (2017), the number of fragments per collision is associated with the number density of breakable asperities, asperity–fragility coefficient, particle surface area, initial value of collision kinetic energy as well as shape parameter, which indicates that not every collision can lead to fragmentation.

[Figure]

Figure R19: The mixing ratio of (a) and (b) graupel/hail, (c) and (d) ice/snow, (e) and (f) rain and (g) and (h) cloud droplet of ice-ice collisional breakup process for different rimed fraction. The rimed fraction of the left column is 0.2 and that of right column is 0.4.

[Figure]

Figure R20: The same as Fig. R19, but for number concentration.

[Figure]

Figure R21: The charging density in IC experiment for different rimed fraction. The rimed fraction of the left column is 0.2 and that of right column is 0.4.

[Figure]

Figure R22: The total charge density in IC experiment for different rimed fractions. The rimed fraction of (a) is 0.2 and that of (b) is 0.4.

References:

Böhm, J. P.: A general hydrodynamic theory for mixed-phase microphysics. Part II: collision kernels for coalescence. Atmos. Res., 27, 275-290, 1992a.

Böhm, J. P.: A general hydrodynamic theory for mixed-phase microphysics. Part III: Riming and aggregation. Atmos. Res., 28, 103-123, 1992b.

Khain., A, Sednev, I.: Simulation of precipitation formation in the Eastern Mediterranean coastal zone using a spectral microphysics cloud ensemble model. Atmos. Res., 43, 77-110, 1996.

Khain, A., M. Pinsky, M. Shapiro, and A. Pokrovsky: Collision Rate of Small Graupel and Water Drops. J. Atmos. Sci., 58, 2571–2595, https://doi.org/10.1175/1520-0469(2001)058<2571:CROSGA>2.0.CO;2, 2001.

Khain, A., Leung, L. R., Lynn, B., and Ghan, S.: Effects of aerosols on the dynamics and microphysics of squall lines simulated by spectral bin and bulk parameterization schemes. Journal of Geophysical Research, 114(D22), D22203, https://doi.org/10.1029/2009JD011902, 2009.

Phillips, Vaughan, T. J., Yano, Jun-Ichi, Khain, and Alexander: Ice Multiplication by Breakup in Ice-Ice Collisions. Part I: Theoretical Formulation. Journal of the Atmospheric Sciences, 2017.

- Please provide more details about the collisions considered in 'mode 1' of the Phillips et al. (2018) parameterization. Were collisions with ice nucleating particles (INPs) other than small ice particles taken into account? A brief description of the primary ice production mechanisms encompassed within the scheme would also be useful.

Reply: Thank you for your professional comment. The "model 1" collision represents the collision between frozen drops and smaller ice crystals. The collisions with ice nucleating particles (INPs) are not considered in this SIP process. The default primary ice nucleation parameterizations implemented in SBM in WRF are used. The immersion freezing is parametrized according to Bigg (1953). The deposition/condensation nucleation is represented using the parametrization of Meyers et al. (1992), which is a function of saturation ratio with respect to ice. The contact freezing is also developed in Meyers et al. (1992), which is a function of temperature.

References:

Bigg, E. K. : The formation of atmospheric ice crystals by the freezing of droplets. Q. J. R. Meteorol. Soc., 79(342), 510–519, doi:10.1002/ qj.49707934207, 1953.

Meyers, M. P., P. J. DeMott, and W. R. Cotton: New primary ice nucleation parameterizations in an explicit cloud model. Journal of Applied Meteorology., 31, 708– 721, doi:10.1175/1520-0450(1992)031<0708:NPINPI>2.0.CO;2, 1992.

3. To improve readability, please consider incorporating a dedicated paragraph (for example in Section 2) that outlines the various measurements utilized in this study, discussing any uncertainties and/or any post-processing applied to them. This applies to the radar observations (Figure 3), sounding data (Figure2) as well as the observed flash rates (Figure 6). Consider moving the information about the lightning observational dataset from the "Results" section (Lines 194-197) to the corresponding data paragraph.

Reply: Thank you for your comment. The following descriptions are added in the paper: *"Radar reflectivity can be used to illustrate the intensity of the storm. The radar data used in this study is a gridded product generated based on 32 S-band radars operated across southeast China. For each radar, the detection radius is 230 km, the range resolution is 250 m and the beamwidth is 1°. The radar finishes a volume scan every 6 minutes consisting of 9 elevation angles (0.5°, 1.5°, 2.4°, 3.4°, 4.3°, 6.0°, 9.9°, 14.6° and 19.5°). The data recorded by these radars were interpolated into a Cartesian grid with a horizontal resolution of 1 km and vertical resolution of 500 m based on the Cressman technique.*

*In addition, the lightning location and flash rate is evaluated using observation. The lightning location data is obtained based on the very low frequency (VLF) lightning location network (LLN) in China developed by Nanjing University of Information Science and Technology (Li et al., 2022). VLF-LLN was established in 2021 and has 26 stations distributed across various regions in China. The detection area covers the entire China as well as parts of East Asia and Southeast Asia. The lightning location method is developed based on the time-of-arrival (TOA) method, and the arrival times of each lighting-induced pulse at different stations are obtained by matching the recorded waveforms to the idealized waveform established using the Finite Difference Time-Domain (FDTD) technique. The lightning location error is 1-5 km.*

*Moreover, the NCEP reanalysis data is used to investigate the synoptic conditions, the sounding measurements at Fuyang, which is conducted every 12 hours, is used to investigate the thermodynamic conditions, and the brightness temperature (TBB) on FY2H satellite that is developed in China is used to illustrate the cloud coverage.*

4. Please explain how the modeled composite reflectivity (shown in Figure 5) is derived. Which parameters (e.g., mass and concentration of ice and liquid hydrometeors) have the most influence on simulated reflectivity? In this way, the reader can better understand the changes caused when SIP is accounted for and you can better support your statement in Lines 251-252 "…the decrease in the sizes of these solid particles is probably the main reason of the weaker composite radar reflectivity in the 3SIP experiment".

Reply: Thank you for your comment. For each grid point, the maximum radar reflectivity among all layers is used as the composite reflectivity. The size of particles has the most influence on simulated reflectivity, which is calculated for a wavelength of 10 cm.

5. For improved visual comparison between model simulations (Figure 5) and radar observations (Figure 3), you may consider including all relevant subplots into a single figure. Also, ensure consistency in colorbar limits (dBZ) across visualizations.

Reply: Thank you for your comment. The subplots of simulated and observed radar reflectivity with the same color bar have been combined in the revised manuscript.

[Figure]

Figure R23: The simulated radar reflectivity and observed reflectivity. (a, b, c)

simulated reflectivity of the experiment without SIP process, (d, e, f) simulated reflectivity of experiment with four SIP processes, (g, h, i) observed reflectivity. The black horizontal line in (a) shows where the cross sections were made.

6. Lines 176-178: Here the reader is already wondering why activating SIP in the model leads to reduced modeled reflectivity. You could mention that this aspect will be elaborated upon in Section 3.2.

Reply: Thank you for your comment. According to your suggestion, a sentence is added to indicate this aspect will be elaborated upon in Section 3.2.

7. Lines 180-181: "…the simulation with all the three SIP processes has the best performance comparing to the observation (Figs. 3b and 5j) ". The robustness of this statement can be enhanced by including additional statistics to complement the visual comparison.

Reply: We appreciate your comment. According to this comment and a comment by another reviewer, we plot the contoured-frequency-by-altitude diagram (CFAD) of reflectivity, which can statistically show the difference in the reflectivity at different heights between observation and model simulations. According to your comment, ice sublimational breakup has been added to our model as the fourth secondary ice production mechanism (Deshmukh et al., 2022; Waman et al., 2022). The experiment with all four SIP processes included is named "4SIP". As seen in Fig. R24, the maximum reflectivity is observed at about 4 km, which is height of the melting levels. The modeled maximum reflectivity from noSIP experiment is larger than observed by about 7 dBZ, this is also seen from the map of composite reflectivity in the paper. With SIP implemented, the maximum reflectivity decreases and is more consistent with observation. The mean reflectivity profiles in both the noSIP and 4SIP experiments are systematically larger than observed as the occurrence frequency of reflectivity greater than 30 dBZ is higher, but the 4SIP performs better than noSIP experiment. The observed reflectivity maybe underestimated at low levels because the lowest elevation angle used in the radar measurement is 0.5 degree (please see more information of measurements in reply to comment 3) and the low-elevation beams are affected by ground clutters (Fig. R25).

[Figure]

Figure R24. The contoured-frequency-by-altitude diagram (CFAD) of reflectivity from (a-c) noSIP, (d-f) 4SIP experiments and (g-i) radar observation. The black lines indicate the profiles of mean reflectivity, and the dashed and dotted lines in (g-i) are the mean reflectivity profiles from noSIP and 4SIP experiments.

[Figure]

Figure R25. Observed radar reflectivity at 500m and 1000m a.m.s.l at 02:00, Nov. 28th.

8. Line 212: consider using a more suitable transition sentence, especially since the charge structure will not be discussed in Section 3.2.

Reply: Thank you for your comment. According to your suggestion, the transition

sentence has been revised.

*"The various SIP processes may have different impacts on the cloud microphysics."*

9. Line 218: Please clarify the meaning of "strong correlation" in this sentence.

Reply: Thank you for your comment. The temporal evolution of the rain mixing ratio is consistent with that of snow, suggesting the melting of snow contributes significantly to the rain. This sentence has been revised in the paper.

10. With the model you have access to all production rates of important microphysical processes, like riming, aggregation, sedimentation, or the melting of graupel particles or snowflakes that could be used to support your statements throughout the text, such as Lines 218, 222, 259, 285, and 287.

Reply: Thank you for your comment. A simple melting procedure is used in the Fast-SBM model, which means that all ice-phase particles simply melt into water. This process is not a complex process carried out by particles at certain scales, but rather a simple transformation of the mass of ice-phase particles to liquid particles. Therefore, it is not sensitive to environmental parameters. The production rate of rime process, aggregation process and sedimentation process are shown in Fig. R26. The rime process occurs mainly between -10°C and 0°C, and the aggregation process occurs mainly in colder regions. The production rate of rime process is greater than that of aggregation process.

[Figure]

Figure R26: The production rate of (a) rime process, (b) aggregation process and (c) sedimentation process.

11. Line 224: Are you referring to the 'riming of cloud droplets and raindrops' rather than the 'rime-splintering process' here? Indeed, liquid hydrometeors that rime onto graupel would typically increase its mass. However, if RS is activated, part of this rimed mass would then be transferred from the graupel to the smaller cloud ice particles.

Reply: Thank you for your comment. The graupel mixing ratio is enhanced in the RS experiment compared to the noSIP experiment, indicating the secondary ice produced by rime-splintering enhances the riming in the cloud. This sentence is revised accordingly.

12. Line 229-230: Any idea why the enhancement of graupel/hail and ice/snow is followed by an increase in the cloud liquid water content (rain + cloud mass mixing ratios)? I would expect the opposite behavior, because of the Wegener–Bergeron–Findeisen (WBF) process.

Reply: Thank you for your comment. The old Fig. 7 shows the domain average mixing ratio, which may be not suitable for investigation. Now we average the mixing ratio only in the cloud. According to the new results (Fig. R27), the rime-splintering and shattering of freezing drops enhance the graupel and snow mixing ratio. The ice-ice collisional breakup and snow breakup during sublimation enhance the ice/snow mixing ratio after 00:00, mainly above 6 km. We do see a decrease in LWC at temperatures colder than 0 °C after adding rime-splintering and shattering of freezing drops. In some areas below the melting level, the LWC may increase due to the enhanced snow concentration that fall from above. The enhanced snow concentration and mixing ratio by SIP does not provide stronger rain, due to their smaller sizes. This has been clarified in the paper.

13. Figure 7: I would suggest superimposing the isotherms in this plot for better visualization of the RS temperature zone, melting layer, and temperatures where IC and SD are efficient.

Reply: Thank you for your comment. The isotherms have been superimposed in the figure in the revised paper.

14. Figure 8: Please explain how averaged concentrations were calculated. Did you consider only in-cloud conditions? Instead of having separate plots for the number

concentrations and sizes, it might be worth plotting the particle size distributions (PSDs) (i.e., d(N)/d(logD)). In this way, the reader would more easily identify both the ice enhancement caused when SIP is considered in the simulations, and the shift of the PSDs towards smaller sizes, which is crucial for capturing the correct radar reflectivity values.

Reply: The number concentrations shown in old Fig. 8 are the whole inner domain average value. In the revised paper, we average the concentration only in the cloud. We agree that PSDs can better show the shift of the PSDs towards smaller sizes. But unfortunately, since we use SBM, there are 132 3D variables in the PSDs of different hydrometeor species, this requires extensively more computer storage and much more cost for the data. For this reason, we only show the concentration and diameters.

15. The discussion of Figure 8 in the last paragraph of Section 3.2, should be more quantitative. You mention that SIP processes can "slightly enhance" or "slightly decrease" the ice-particle or liquid-particle concentrations, respectively. Please try to quantify the ice enhancement caused when SIP is included in the model compared to the noSIP sensitivity simulation. This is an important information if you want to convince the reader of the importance of incorporating SIP processes in the model.

Reply: Thank you for your comment. To provide better quantitative analysis, we plot the difference in mixing ratio and concentration between the simulation with SIP implemented and the noSIP simulation (Fig. R27 and R28). The graupel and ice/snow concentration are enhanced by rime splintering and shattering of freezing drops, mainly below 8km. The maximum increase, which exceeds 0.02 g/kg, is found between 00:00 and 04:00. If all four SIP processes work together, the ice/snow concentration is clearly higher than that without SIP. The rain concentration above the freezing level decreases due to the four SIP processes, suggesting a more significant cloud glaciation by SIP processes. These quantification results are added in the revised paper.

[Figure]

Figure R27: Differences in the mixing ratio of different hydrometeors between the experiments with SIP and that without SIP. (a, f, k, p) experiment with rime-splintering, (b, g, l, q), experiment with ice-ice collisional breakup (c, h, m, r) experiment with shattering of freezing drops, (d, i, n, s) experiment with ice breakup during sublimation, and (e, j, o, t) experiment with four SIP processes.

[Figure]

Figure R28: The same as Fig. R27, but for number concentration.

16. In Section 3.2 or the "Discussion and Conclusions" section, consider including a discussion on the relative contribution of each SIP mechanism and a comparison of your findings with similar convective case studies from the literature.

Reply: Thank you for your comment. The following discussion is added to the revised paper.

"*Different SIP processes have different impacts on the cloud microphysics electrification. The rime-splintering and shattering of freezing drops are active throughout the cloud life cycle but are limited to relatively warm temperatures. The cloud glaciation below 8 km is enhanced by these two processes, leading to lower LWC at higher levels. The low-level positive charging is significantly enhanced by them due to the higher graupel and ice/snow concentrations. The ice-ice collisional breakup is more active in regions with higher ice/snow mixing ratio, its average impact on cloud electrification is minor, while it could be significant in some areas in the cloud. The sublimational breakup of snow is more active near cloud edges or in downdrafts, and its average impact on cloud electrification is weak.*

*Due to the scarcity of winter thunderstorms, there have been few modeling studies of it. Takahashi et al. (2019) studied the winter clouds in Hokuriku and found that lightning was generated in clouds with the following conditions: cloud top temperature less than -14°C, -10 °C isotherm is higher than 1.2 km, space charge greater than 2-3 pC/L, ice crystal concentration greater than 500 m$^{-3}$, and graupel concentration greater than 20 m$^{-3}$. According to the analysis above, the winter thundercloud studied in this paper satisfies all these characteristics. Takahashi et. al. (2017) pointed out that winter thunderstorm clouds have lower LWC and low cloud tops. In our simulation, the modeled LWC is typically lower than 1 g m$^{-3}$, which is lower than that reported in summer convections (e.g., Yang et al., 2016; Phillips et al. 2022). The lower LWC in wintertime convection indicates weaker riming, thus a lower riming accretion rate, which potentially leads to a higher possibility of inverted charge structure of thunderstorms (Wang et al. 2021).*"

References:

Takahashi, T., Sugimoto, S., Kawano, T., and Suzuki, K.: Microphysical structure and lightning initiation in Hokuriku winter clouds. Journal of Geophysical Research:

Atmospheres, 124, 13,156–13,181, https://doi.org/10.1029/2018JD030227, 2019. https://doi.org/10.1016/j.atmosres.2015.05.006, 2015.

Takahashi, T., Sugimoto, S., Kawano, T., and Suzuki, K.: Riming Electrification in Hokuriku Winter Clouds and Comparison with Laboratory Observations. Journal of the Atmospheric Sciences, 74(2), 431–447, https://doi.org/10.1175/JAS-D-16-0154.1, 2017.

Yang, J., Wang, Z., Heymsfield, A., and Luo, T.: Liquid-ice mass partition in tropical maritime convective clouds. J. Atmos. Sci., 73, 4959-4978, doi: 10.1175/JAS-D-15-0145.1, 2016.

Wang, D., Zheng, D., Wu, T., and Takagi, N.: Winter Positive Cloud‐to‐Ground Lightning Flashes Observed by LMA in Japan. IEEJ Transactions on Electrical and Electronic Engineering, 16(3), 402–411, https://doi.org/10.1002/tee.23310, 2021.

17. Line 383-384: The transition sentence does not have a clear connection with the rest of the paragraph.

Reply: Thank you for your comment. We agree. In addition, this sentence is kind of repeating the rest of the paragraph. Therefore, it is removed in the revised paper.

18. Line 421: You may want to refer to the new empirical parameterization for the sublimational break-up mechanism developed in Deshmukh et al. (2022). This mechanism has been found to be the second most dominant SIP mechanism in fast convective downdrafts (Waman et al., 2022).

Reply: Thank you for your comment. The sublimational breakup mechanism has been added in our model and its impact has been discussed in manuscript. All the related figures are updated. The results show the sublimational breakup of ice is more active near cloud edges or in downdrafts. On average, its impact on cloud electrification is weaker than the rime-splintering and shattering of freezing drops, but it could be significant in some areas.

Technical corrections:

• Line 124: I would suggest "grid spacing" instead of "resolution"

Reply: "resolution" is changed to "grid spacing".

• Line 169: I would suggest "Model evaluation" instead of "Model validation"

Reply: "Model validation" is changed to "Model evaluation".

• Line 156: "correct representation" (not representative)

Reply: "representative" is changed to "representation".

• Line 228: "shown later", consider indicating the section where the subsequent discussion will take place. Similarly, for Line 233.

Reply: Revised accordingly.

• Line 233: "reduced by SIP" (not by this SIP)

Reply: "reduced by this SIP" is changed to "reduced by SIP".

• Line 257: Section 3.3 (not 3.2)

Reply: "Section 3.2" is changed to "Section 3.3"

• Line 361: Section 4 Discussion and Conclusions (not 5)

Reply: "Section 5" is changed to "Section 4".

• Line 371: suggests (not suggest)

Reply: "suggest" is changed to "suggests".

• Please double check the reference provided for Mansell et al. (2010)

Reply: The reference is revised to "Mansell, E. R., Ziegler, C. L., and Bruning, E. C.: Simulated electrification of a small thunderstorm with two-moment bulk microphysics. J. Atmos. Sci., 67, 171–194, doi: 10.1175/2009JAS2965.1, 2010."

---

## Author Response (AR2)

Review for: Impact of ice multiplication on the cloud electrification of a cold-season thunderstorm: a numerical case study, by Yang et al.

The reviewer thanks the authors for their diligent efforts in addressing the comments. The revised manuscript has significantly improved, and only a few minor suggestions are outlined below:

1. A general comment is that the revised manuscript now includes 21 figures, which makes it challenging for readers to focus on the key takeaways. I recommend considering the relocation of certain figures – particularly those not central to the paper or discussed minimally in the manuscript (e.g., Figure 1, 2, 11, 12) – to the Supplementary Material in order to enhance clarity.

Reply: We appreciate your comment. According to this comment and comment 3, Figures 11 and 12 have been removed from the manuscript. Now there are 19 figures. Figures 1 and 2 show in detail the weather conditions of this winter thundercloud, we prefer to keep them in the text because we think they are important for case description.

2. In line 230 you mention "the good performance of WRF": considering the observed discrepancy between model and observations illustrated in Figure 5, using a phrase like "composite reflectivity is simulated reasonably well" might be more accurate.

Reply: Thank you for your comment. "The good performance of WRF in modeling the composite reflectivity and the improvements by SIP provide us the confidence to investigate …" has been changed to "Based on the facts that composite reflectivity is simulated reasonably well and the SIP processes result in improvements, we are confident to investigate …".

3. Related to the comment #1 raised above, I am not sure whether Figs. 11 and 12 contribute significantly to the paper, especially considering the absence of measurements for comparison. Figs. 8 and 9 seem sufficient to discuss the WRF sensitivity to various SIP processes. Additionally, the production rates of SIP processes presented in Fig. 13 seem more valuable for interpreting the observed ice enhancement than Figs. 11 and 12.

Reply: We agree. Figures 11 and 12 have been removed from the manuscript, and the related text are revised.

4. When activating all SIP mechanisms in the 4SIP sensitivity simulation, you mention that the ice enhancement "maybe weaker than the impact of a single SIP process" (Lines

313-314). Why do you think this happens? I propose considering the addition of a subplot in Fig. 13 to illustrate the synergistic impact of all SIP processes in the 4SIP simulation. Contour lines can be used to indicate with different colors when each SIP mechanism included in 4SIP surpasses a predefined threshold (for example: 0.1 #/L/s for rime splintering, 0.01 #/L/s for droplet shattering or collisional break-up and 10(-4) for sublimational break-up). This subplot could reveal whether one SIP process dominates, potentially reducing the cloud liquid water content and thereby diminishing the impact of the remaining mechanisms.

Reply: Thank you for your comment. The figure is updated accordingly, which shows the temporal variation of the mean ice production rate by different SIP mechanisms. By using the thresholds you suggested, the figure looks messy, thus we use a single threshold of $3 \times 10^{-3}$ $L^{-1}s^{-1}$. The results show the rime-splintering dominates the secondary ice production between 0 and -10 C, potentially reducing the cloud liquid water content and thereby diminishing the impact of the remaining mechanisms. The ice production rate by sublimational breakup is so small that it never meets the threshold.

[Figure]

Figure R1: Cross-sections of the secondary ice production rates by different SIP processes resulting from the 4SIP experiment at 01:00 Nov. 28th. (a) rime-splintering, (b) ice-ice collisional breakup, (c) shattering of freezing drops, (d) sublimational breakup of ice, and (e) the time-height diagram of the mean ice

production rate by different SIP processes. Contour levels are $3\times10^{-3}$, $5\times10^{-3}$, $10\times10^{-3}$, $20\times10^{-3}$, and $30\times10^{-3}$ $L^{-1}s^{-1}$, the ice production rate of sublimational breakup of ice is so small that it never meets the lowest contour level

5. In your manuscript, it is noted that a rimed fraction of 0.2 was prescribed in the IC and 4SIP experiments, with the efficiency of the ice-ice collisional break-up process being sensitive to the choice of this parameter (Lines 543-544). I would suggest emphasizing this important assumption not only in Section 4 but also in Section 3, particularly when discussing the limited efficiency of the collisional break-up mechanism in comparison to, for example, rime splintering.

Reply: Thank you for your professional comment. This following discussion has been added to Section 3.

*Ice-ice collisional breakup is more intense in regions with high ice/snow concentrations (Fig. 12f, l), its secondary ice production rate is much smaller than that of rime-splintering. However, it should be noted that the efficiency of ice-ice collisional breakup is related to the rimed fraction (Karalis et al., 2022; Sotiropoulou et al., 2021), A sensitivity test shows using a larger rimed fraction (0.4) can result in a stronger impact of ice-ice collisional breakup on cloud microphysics, but it is still much weaker than that of rime-splintering.*